# Understanding Tropical Forest Abiotic Response to Hurricanes using Experimental Manipulations, Field Observations, and Satellite Data

Ashley E. Van Beusekom[1], Grizelle González[1], Sarah Stankavich[2], Jess K. Zimmerman[2], and Alonso Ramírez[3]

[1]USDA Forest Service International Institute of Tropical Forestry, Río Piedras, Puerto Rico 00926, USA
[2]Department of Environmental Sciences, University of Puerto Rico, Río Piedras campus, San Juan, Puerto Rico,00931, USA
[2]Department of Applied Ecology, North Carolina State University, Raleigh, North Carolina 27605, USA

*Correspondence to*: Ashley E. Van Beusekom (AVBscience@gmail.com)

**Abstract.** With projected increasing intensity of hurricanes and large uncertainty in the path of forest recovery from hurricanes, studies are needed to understand the fundamental response of forests to canopy opening and debris deposition: the response of the abiotic factors underneath the canopy. Through two manipulative experiments and instrumenting prior to hurricane María (2017) in the Luquillo Experimental Forest (LEF) of Puerto Rico, this study found a long recovery time of primary abiotic factors (beneath canopy light, throughfall, and temperature) influenced by the disturbance of canopy opening, and complex responses by the secondary abiotic factors (relative humidity, soil moisture, and leaf saturation) influenced by the disturbance of the primary factors. Recovery took 4-5 years for beneath canopy light, while throughfall recovery took 4-9 years and neither had recovered when hurricane María passed 3 years after the second experiment. Air and soil temperature seemingly recovered quickly from each disturbance (<2.5 years in two experiments for ~ +1 °C of change); however, temperature was the most important modulator of secondary factors, which followed the long-term patterns of the throughfall. While the soil remained wetter and relative humidity in the air stayed lower until recovery, leaves in the litter and canopy were wetter and drier, with evidence that leaves dry out faster in low rainfall and saturate faster in high rainfall after disturbance. Comparison of satellite and field data before and after the 2017 hurricanes showed the utility of satellites in expanding the data coverage, but the muted response of the satellite data suggest they measure dense forest as well as thin forest that is not as disturbed by hurricanes. Thus, quick recovery times recorded by satellites should not be assumed representative of all the forest. Data records spanning the multiple manipulative experiments followed by hurricane María in the LEF provide evidence that intermediate hurricane frequency has the most extreme abiotic response (with evidence on almost all abiotic factors tested) versus infrequent or frequent hurricanes.

## 1 Introduction

Hurricanes are expected to increase in intensity with climate change (Emanuel, 1987; Knutson et al., 2010; Yoshida et al., 2017), thus understanding how tropical forests respond to hurricanes is critical to understanding future forest regimes. Tropical

forests are in a cycle of non-equilibrium, a cycle driven by the response to the large step-changes of hurricanes (Burslem et al., 2000). Recently, new tools for understanding the nature and duration of the forest-hurricane response have become available for use; satellite data can provide landscape-wide qualities of the historical response (Schwartz et al., 2017) and earth systems models can provide long-term forest response given the projections of increased frequency of hurricanes (Lee et al.,

2018). While these tools can provide a large amount of spatially-complete, cost-effective, and consistently-recorded data, the data needs to be placed in context of what is actually happening at the ecosystem level. There is a need for connection between disturbance and recovery at the critical forest scale: for the manner in which landscape-scale data downscales to the more critical forest landscapes, and for the measured response of the forest with repeated hurricanes that should be put into a long-term model (Bustamante et al., 2016; Holm et al., 2014). These connections can only be accomplished with the analysis of

fine-scale field observational data. Instead of trying to estimate if and when the vegetation has returned to its pre-disturbance state, insight on ecosystem health can be gained by studying how the abiotic factors respond to the disturbance. Cascading effects due to canopy openness account for most of the shifts in the forest biota and biotic processes (Shiels et al., 2015), and the biotic environment responds to changes in the abiotic environment.

To this end, a manipulative experiment on hurricane disturbance effects was implemented in the Luquillo Experimental Forest (LEF) in northeastern Puerto Rico, with measurements starting January 2003 and continuing through the time of this manuscript. The LEF represents a tropical wet montane forest, with high rainfall, high productivity, frequent hurricane disturbance, and semi-frequent droughts (González et al., 2013; Scatena and Lugo, 1995; Wang et al., 2003). The forest extends from sea level to 1 km peaks. Droughts occur twice as frequently as hurricanes on the island (every 10 and 21 years

respectively) and affect the forest often as dry spells or stronger (Scatena, 1995; Waide et al., 2013). The Canopy Trimming Experiment (CTE) was designed to study the key mechanisms behind a tropical forest's response after a major hurricane, and guide how repeated hurricanes might be expected to alter such ecosystems using these key mechanisms (Richardson et al., 2010; Shiels et al., 2015; Shiels and González, 2014). Multiple control and treated plots were created in the forest. In the treated plots, the forest canopy was trimmed and the canopy debris was littered to the forest floor to simulate the canopy

changes from a category 3 hurricane (on the Saffir Simpson scale).

Two large disturbances occurred during the experiment, both of which were measured by satellites as well as the field instruments. In summer 2015 a drought affected the LEF, starting in May 2015. The forest was still experiencing drought conditions until March 2016 (https://droughtmonitor.unl.edu), although precipitation increased after September 2015. On

September 20, 2017, category 4 hurricane María made a direct hit on the CTE site. A relatively small amount of disturbance was attributed to the offshore passing of hurricane Irma 2 weeks earlier; the CTE site was on the lee of hurricane Irma. The drought and the hurricanes provided data beyond the experimental manipulations on how the forest abiotic environment responds to canopy opening and debris modulated with the variance of climate seasonal cycles and irregularities.

A simplified way of thinking about response to canopy opening and debris deposition is to consider three levels of response. Primary factors are only affected by the initial disturbance: more light and throughfall reach the forest floor and temperatures under the canopy increase. Secondary factors are affected by the primary inputs: relative humidity (in the air), soil moisture,
and leaf saturation (wetness of canopy and litter leaves) levels change under the canopy. Tertiary factors are biotic, which are affected by primary and secondary factors, the abiotic factors. Research on biotic effects of hurricane disturbance are numerous (for synthesis efforts see: Mitchell, 2013; Shiels et al., 2015) but less researched is how the abiotic factors have changed to alter the biotic environment. This study attempts to quantify abiotic response as acute changes from a hurricane disturbance (experimental or otherwise) and recovery from the changes, for primary and secondary factors. Quantifying the responses
makes it possible to assess if the experimental trimming data and satellite data are reasonable sources for studying the effect of hurricane disturbance and appear to be measuring the same abiotic system, as well as appreciate if different events cause substantially different responses. This study does not attempt to determine what amount of recovery is considered 'normal' conditions to biotic life, or in other words what would affect tertiary factors, but instead quantifies changes in the abiotic factors that can be used to frame the changes found in biotic factors post-hurricane in many previous studies including those
of biotic abundance (Shiels et al., 2015), soil biochemistry (Arroyo and Silver, 2018), litter decomposition (González et al., 2014; Lodge et al., 2014), and plant reproduction (Zimmerman et al., 2018).

## 2 Methods

In spring of 2005 (CTE1) and December of 2014 (CTE2), in 0.09 ha square plots near the El Verde Field Station (419 m; 18°20' N, 65°49' W), the forest canopy was trimmed in 3 treatment plots, and the canopy debris was littered to the forest floor.
The plot size and trim amounts were based on the patch disturbance after the two most recent hurricanes before 2017, both category 3 hurricanes at the location of El Verde: Hugo in September 1989, and Georges in September 1998 (Zimmerman et al., 2014). Non-palm trees of substantial size (>15 cm diameter) had their smaller branches (<10 cm diameter) removed. Smaller non-palm trees and all palm trees were trimmed at 3 m height. All the trimming debris was added to the plot from which it was obtained from, with the debris in each plot supplemented with outside debris if necessary, to keep the amounts
and kinds of debris equal across the plots.

Biotic and abiotic data were collected in the inner 0.04 ha quadrants of the 0.09 ha trimmed plots to minimize edge effects. Details of the biotic responses to the 2005 experiment have been extensively documented (Richardson et al., 2010; Shiels et al., 2014, 2015; Shiels and González, 2014), but the abiotic responses (after CTE1 or CTE2) were not fully analysed until now.
Primary factor data (beneath canopy light, throughfall, and air and soil temperature) and secondary factor data (air relative humidity, soil moisture, and leaf saturation) were collected in all plots. To account for spatial heterogeneity under the canopy,

multiple sensors and measurements in each plot were used and the results are calculated off of data averaged from all control and treated plots (with quality control).

While the CTE1 and CTE2 data were being collected, abiotic data were also being collected by satellites and by a nearby weather station. The weather station was located on a tower 30 m above the ground, above control (untrimmed) canopy. After hurricane María, comparisons could be drawn between the experiment and the actual hurricane response, as well as an analysis of which aspects of the response were captured by satellite data, MODIS and AMSR2. It is important to note that hurricane María provided a much larger hurricane trimming effect than the CTEs were designed to simulate.

**2.1 Collecting and Homogenizing Time Series Data Types**

Abiotic field data after 2015 were collected sub-hourly by automated sensors and averaged into daily values. The abiotic field data before 2015 were collected by different sensors or more intermittent methods (soil and litter gravimetric water contents (GWCs) and canopy photos), so the data had to be converted and calibrated from this first period to the post-2015 period in order to make one time series. Satellite data also had to be converted and calibrated to the post-2015 data type. Specific methods

of collection, conversion, and calibration of each data type will be detailed in the following subsections.

Many of the data types required calculation of a smoothed data pattern in order to convert and calibrate. In all cases, the smoothing was done using Local Estimated Scatterplot Smoothing (LOESS), which fits least squares polynomials locally to the points. The LOESS degree of smoothing is contingent on the size of the local neighborhood, which here was always chosen

to be one year of data around each point. The yearly smoothing was done to extract the larger signal from the data and to homogenize the different collection intervals of the data. The automated sensor field data captured larger amounts of background noise than the temporally smoothed rain funnel data and the geographically smoothed satellite data; and to a lesser extent, the geographically smoothed soil and litter GWCs and canopy photo data. The one-year smoothing neighborhood was chosen to be longer than the longest length of time between repeat measurements across all data types and methods. No

smoothing was done across any of the event dates, in CTE treated, control, or satellite data, regardless if the data type was affected by each event. These smoothing breaks were used to keep boundary conditions of the LOESS applications more similar.

**2.1.1 Beneath Canopy Light Data**

Light beneath the canopy was quantified with solar radiation data. Solar radiation data were collected after 2015 by a Campbell

Scientific LI200X pyranometer in each plot measuring 400-1100 nm light from sun plus sky radiation. Earlier estimates of solar radiation were made with sets of hemispherical canopy photos, ten photos in each set in each plot, which were taken approximately every 4 months 2005-2012. Sets of photos were also taken before the first experiment, and once a year 2015-2017. The solar radiation field data were compared to MODIS Aqua and Terra satellite leaf area index (LAI) data at 500 m, 8-

day resolution. The Beer-Lambert law (Monsi, 1953) was used to convert the LAI data into solar radiation estimates, calculating the attenuation the canopy with a specific LAI invokes on the available (above-canopy) light. Annual patterns of photosynthetically active radiation (PAR) extinction coefficients are needed to calculate the attenuation given by the Beer-Lambert law. An annual pattern of these extinction coefficients was solved for by using two years of data of the field-measured CTE2 control plot solar radiation, the tower weather station above-canopy solar radiation, and the MODIS LAI data. The three sets data were interpolated or averaged to daily values, and then the coefficients were calculated on the two years of data before the hurricane (so excluding the 2015 drought). These annual patterns were averaged and smoothed into one annual pattern of extinction coefficients that was applied for every year of the MODIS data.

In the reanalysis of the CTE1 data presented here, canopy photos were converted to global solar radiation data with a modified version of the Hemiphot method (ter Steege, 2018) as follows. Images were converted from color to black and white with a threshold, where the threshold is found iteratively for the best separation of background and foreground using the Ridler and Calvard method (Bachelot, 2016); this method requires calibration. Thresholding was later calibrated to have agreement between annual patterns in the photo solar radiation data and annual patterns in the instrument measured solar radiation data measured in the control plots. Next, the black and white images were converted to canopy openness data by calculating openness on concentric rings of the photo representing sky hemisphere with an arc of 1 degree.

Then, PAR was calculated under the canopy for every day of the year before and after each photo, assuming a constant canopy cover for those time periods. The PAR was then made into one daily time series at each photo site by linearly interpolating PAR each day as a fraction of the previous and the next photo's calculated PAR on that day. This roughly interpolated the canopy cover changes due to recovery from the trimming, and interpolated seasonal changes in canopy cover as long as the photos were repeated every winter and summer.

The PAR is the sum of direct and indirect light. The direct light was calculated from the path length of the sun's light through the atmosphere to the forest and the atmospheric transmissivity. Atmospheric transmissivity was given variability around the standard tropical value assuming a linear relationship with relative humidity in the air (Winslow et al., 2001) (as measured above canopy). Path length was calculated from the sun's orbital position on each day of the year relative to the forest. Diffuse light was calculated assuming each part of the sky is equally bright and thus diffuse light is a fraction of direct light. Underneath the canopy, PAR can be approximated as the sum of the direct light through all open parts of the canopy and the diffuse light multiplied by 15% (based on empirical equations; Gates, 2012). Global solar radiation is then approximated as a multiple of PAR (2X; see Escobedo et al., 2009) calibrated to solar radiation measurements from above the canopy at the tower weather station.

### 2.1.2 Throughfall Data

Throughfall data were collected the entire time period with the same method of bi-weekly recordings of rain funnels. These funnels were 9.2 cm in diameter, with 1000 $mm^3$ volume. Throughfall was also collected in the treated plots sub-hourly after 2015 with automatic rain gages. The rain gage data that overlapped the rain funnel data was used to calibrate the rain funnel data.

### 2.1.3 Temperature Data

Temperature data were collected after 2015 by a Decagon Devices VP-3 sensor in each plot in the air 2 m up from the ground and a 5TM sensor in each plot in the soil 0.05 m down into the ground. Earlier temperature data were collected hourly by a Campbell Scientific 107 sensor in each plot in the air and one in the soil, underneath the canopy. Air temperature above the canopy at the tower weather station was calculated with the same instrument the entire time period, so annual patterns of the ratios of above-canopy air temperature to below-canopy air and soil temperature were used to calibrate the 107 data. First, the ratios were calculated for two years of VP-3 data before the hurricane (so excluding the 2015 drought). These annual patterns were averaged and smoothed into one annual ratio pattern for air and one for soil. Then an air and soil annual ratio pattern was calculated for the complete years of the 107 data (so excluding 2005-2007) and the above canopy data, and the difference between the ratios were used to make one annual correction each for air and soil that was applied for every year of the 107 data. The air temperature field data were compared directly to MODIS Aqua and Terra satellite land surface temperature (LST) data at 1 km, 8-day resolution. MODIS LST measures energy balance at the land surface, so is not representative of air temperature under the canopy but it will be affected by changes in air temperature. Annual maximums of LST and air temperature are highly correlated across the globe with correlation strongest in forested areas (Mildrexler et al., 2011), and LST has been shown to respond to forest cover changes in other areas of the tropics (van Leeuwen et al., 2011).

### 2.1.4 Air Relative Humidity Data

Air relative humidity data were only collected after 2015. They were collected by the same Decagon Devices VP-3 sensors in each plot (2 m up from the ground) that collected air temperature. Because only one kind of instrument collected this data, no conversion was done on this data.

### 2.1.5 Soil Moisture Data

Soil volumetric water content (VWC) data were collected after 2015 by reflectometers, with one Decagon Devices 5TM sensor in each plot measuring shallowly at 5 cm deep and three Campbell Scientific CS616 sensors in each plot collecting profiles from the surface to 15 cm deep. The VWC profile data are comparable to measurements of soil moisture collected by drying out soil samples. Such soil samples were collected for GWCs approximately every 3 months 2003-2006, and in 2015, with 5 in each plot. Some of these soil GWCs have been published previously before this reanalysis (Richardson et al., 2010). Here,

soil GWCs were converted to soil VWCs estimates with measurements of soil bulk density recorded at the same time as the GWCs; or, using average values from each plot if direct measurements were not available. The 2015 overlap period between the smoothed data of the sensors and the soil sample data was used to calibrate the converted data. The shallow soil VWC field data were compared directly to AMSR2 descending and ascending track satellite soil VWC data, at 10 km, 1-day resolution.

### 2.1.6 Leaf Saturation Data

Leaf saturation data were collected after 2015 by three Decagon Devices dielectric leaf wetness sensors in the low canopy leaves in each plot 5 m up from the ground and three in the litter leaf layer in each plot. These sensors have similar thermal mass and radiative properties to real leaves, and wetness is measured by the voltage signal output after voltage excitation, which is higher in proportion to the volume of water on the sensor. This voltage output was then assigned 0% saturation (dry) at the lowest recorded value, and 100% saturation at the highest recorded value. Earlier measurements of litter saturation were made with leaf GWC values from litterbags, 5 in each control plot and 10 in each treated plot. These litterbags were made of air-dried, pre-weighed leaves, placed in the litter layer immediately after the CTE1 trimming and retrieved for collection approximately every 3 months 2005-2006. This data was published previously (Richardson et al., 2010). The litterbag procedure was repeated for the CTE2 trimming, and four litterbag measurements of GWC were made in 2015. Leaf GWC is proportional to leaf VWC if the assumption of constant leaf bulk density across plots is made. Then the early litterbag data could be converted to saturation percentages using the ratio between the data of the 2015 litterbags and the smoothed data of the dielectric leaf wetness sensors collected at the same time.

### 2.2 Quantifying Abiotic Interaction and Response

To explore the relationships between primary and secondary abiotic factors, daily means were correlated. All abiotic data after January 2015 were prewhitened by filtering with an autoregressive integrated moving average model (ARIMA; Box et al., 2015) and first-differencing to remove seasonality and trends. The resulting prewhitened data were examined for correlation between primary and secondary factors for periods with daily data (i.e., after CTE2 and continuing after the hurricanes until 2019).

To explore the differences between responses of different abiotic factors, a smooth time series of each factor was computed, as well as annual averages of data starting after each disturbance event and continuing until the next disturbance event. This was done for CTE data as well as satellite and tower weather station data. For the smooth time series, one-year LOESS neighborhoods were used to reduce the noise in the data and extract the larger signal with seasonality. For the annual time series, averages of every 365 days after an event were computed (e.g., after hurricane Maria on September 20, 2017; an average was computed from September 21, 2017 to September 20, 2018). Each yearly mean was visualized as a point at the midpoint of each calendar year (July 1), regardless of the starting date of the average, so that the connected annual time series did not

change visually in its seasonal relationship to the smooth time series throughout the series of disturbances. Thus, the first point after an event represents the average of day 1 to day 365 (year one), the second point the average of year two, and so on.

Acute change of each factor was quantified by pre-defined metrics on the time series. The acute change after the hurricane was defined as the change in the control time series or the satellite time series from right before the hurricane to right after the hurricane, September 20, 2017. The acute change after an experiment disturbance event was defined as the maximum difference between the treated and control time series (in relation to the control time series) on any day between the last day of the canopy trimming (spring 2005, December 2014) and of the next September 20 (year 2005 and 2015, respectively), so

that the experimental changes could be compared to the hurricane changes. These changes were calculated on the smoothed data, in order to reduce the noise in the data but still account for short-term changes that yearly means would not capture.

Recovery after a CTE experiment was defined as the point in time after the acute change that the value of the treated data time series is 'close' to the value of the control data time series, and afterwards the difference between the treated and control data

stayed 'small'. While this definition is qualitatively intuitive, there a several choices in that need to be made to enact it in a quantitative manner. The recovery length is sensitive to the choice of metric for initial closeness and for post small-differences. The recovery length is also sensitive to the summary choice of the raw data and the length of time the data needs to be close to be called 'recovered'.

A sensitivity study was performed to explore the effects of the choices on the reported recovery length. Differences in timeseries were calculated as fractions of the acute change, or

$$\delta(x) = \left. \frac{\frac{T^*(x) - C^*(x)}{C^*(x)}}{\frac{T(x_a) - C(x_a)}{C(x_a)}} \right. \tag{1}$$

where $x$ is the day the data was measured on, $T(x)$ is the treated data value, $C(x)$ is the control data value, and $x_a$ is the day the acute change was measured on. The timeseries of $T$ and $C$ are the smooth LOESS timeseries used to calculate the acute change,

but $T^*$ and $C^*$ were used as the daily (averaged from raw) data timeseries, the smooth LOESS timeseries, and a mean timeseries on the daily data (using 6 month means after each day $x$). Then, the day of recovery was defined as the day $x_r \geq x_a$, where

$$\delta(x_r) \leq b_{\text{init}} \tag{2}$$
$$\max(\delta(X_r)) \leq b_{\text{post}} \tag{3}$$

using days post acute change as $X_r = x_r + 1, x_r + 2, x_r + 3, \dots x_r + n$. In order to capture possible seasonal differences in the

post-period, the ending day for this period was chosen as $n = 182$, or until the next event occurred. Lengths up to $n = 365$ were tested, but the longer period did not make much difference and means the results of the CTE2 (only 2.8 years) have the full post period for substantially less time. The recovery time will be reported as time since the start of the experiment, $x_0$, where $x_0$ is January 1, 2005, for CTE1, and December 1, 2014, for CTE2.

With the definition set in Equations 1-3, the entire recovery solution space could be studied in the sensitivity study using buffer values 0 to 1. If both buffers are 1, the recovery time will be calculated as $x_r = x_a$, and if both the buffers are 0, there will only be recovery if the treated and control timeseries are identical after day $x_r$. The entire recovery solution space was calculated for three different data summary choices as discussed above: daily, smooth (LOESS), and mean. For Equation 2, the mean timeseries is not used as an option, as it would be the same as using Equation 3 with the mean timeseries. Thus, the sensitivity

equations can be run for six scenarios. While all scenarios were tested, results here focus on the end-members of the least summarized: daily data timeseries for both Equations 2 and 3; the most summarized: smooth and mean timeseries in Equation 2 and 3, respectively; and the intermediate: smooth data timeseries for both Equations 2 and 3. Based on the sensitivity study, buffers in the final recovery calculations were chosen to be small, but not too small.

Note, other studies have defined recovery as the year in which the annual maximum value (of the disturbed area) returns to a previous annual maximum value (assumed representative of undisturbed conditions; Lin et al., 2017). While the method used here is dependent on the size of the amount of data smoothing and the size of the buffers; it is able to make use of the parallelly collected control data to calculate more precise recovery lengths than a year. Furthermore, in a frequently disturbed regime such as the LEF, it is difficult to say what year would be representative of undisturbed conditions. Also, this study does not

attempt to quantify the biotic perception of disturbance and just focuses on the abiotic effects, as different biotic species will perceive abiotic disturbance at different sizes of buffers.

## 3 Results

### 3.1 Homogenized Time Series

Some of the secondary factors correlated with the primary factors of solar radiation and temperature, but there were no monotonic relationships found with throughfall. The prewhitened (seasonality and trends removed) air relative humidity correlated well with both the prewhitened primary factors of solar radiation and air temperature ($R^2 = -0.67$) across all periods (after CTE2 and after the hurricanes) and all plots (control and treated). The prewhitened leaf saturation (canopy and litter) correlated somewhat with both the prewhitened primary factors of solar radiation ($R^2 = -0.35$) and air temperature ($R^2 = -0.49$)

across all periods and plots. The prewhitened soil moisture (shallow and profiles) did not correlate consistently well with any of the primary factors. All significant correlations were highest at zero lags.

The smoothed time series allowed calculation of more detailed responses than if the analysis had been restricted to only calculations on annual averages. The CTE and satellite acute changes after each disturbance event as calculated from the

metrics on the smoothed time series (vertical bars on Figures 1, 2 and Table 1) are much larger than what was seen with the

annual averages of the data (differences in dashed lines on Figures 1, 2). Throughfall is the only exception to this and its acute changes could be accurately summarized with the annual average changes (Figure 1b).

## 3.2 Sensitivity Studies on Calculated Recovery Times

The sensitivity studies reflected the trade-offs in recovery time calculation between extracting the seasonal data signal and capturing the daily variation. The more summarized the timeseries, smooth signals of control and treated data result and the sooner post-treatment the abiotic factor was calculated as having small differences for an entire 6 months (and thus satisfied the post buffer size requirement for recovery). However, with the lesser amount of daily variation in the more summarized timeseries, the control and treated data do not approach each other on any one day (and thus do not satisfy the initial-closeness

buffer size requirement for recovery) until a long period of the two timeseries is very similar. Conversely, the less summarized the timeseries, the more daily variation (or noise from an idealized signal) appears in the reported time series and the opposite situation occurred. Figures 3 and 4 show that when the buffers are large, the daily data sensitivity surfaces plot on top (longer recoveries) and the smooth-mean data sensitivity surfaces plot on bottom (shorter recoveries). When the initial closeness buffer gets small, the daily data is quite often not calculated as recovering during the time of the experiment. When the post small-

differences buffer gets small, the choice of summary for the data in Equation 2 (the initial closeness) mattered more, with the smooth-mean data sensitivity surfaces showing longer recovery than the daily data sensitivity surfaces. The third set of sensitivity surfaces, those from the smooth data, show a reasonable compromise in processing amount versus noise amount.

Because the data noise was seen to overwhelm the recovery calculation on abiotic factors with a smaller acute change signal,

buffers in Equations 1-3 had to be weighted the by the size of the acute change relative to the annual (undisturbed) range, or for each abiotic factor

$$ b = b^* \Big/ \sqrt{\left| \frac{T(x_a) - C(x_a)}{\text{annual\_range}} \right|} \tag{4} $$

for both $b = b_{init}$ and $b = b_{post}$. Then, $b^* = 0$ was used for the Equation 4 calculation of $b_{init}$ and $b^* = 0.15$ was used for the Equation 4 calculation of $b_{post}$. Note, if the data was from an idealized system without noise and not a real system, Equations

1-3 could have been used with the same $b_{init}$ and $b_{post}$ for every abiotic factor. Recovery time calculations from the Equation 4 buffers are in Table 1.

Some of the shortcomings of the data are apparent in the calculations; these affect the accuracy of the resulting recovery times in Table 1. Firstly, abiotic factors with high geospatial variance may have sizable differences even in a recovered state in co-

located plots. This is known issue with throughfall. It is possible that a recovered state for throughfall is observed 4 years after CTE1 (Figure 1b, see the temporary convergence in the annual averages and the smooth data) instead of later as the defined

recovery metrics calculations in Table 1 report. This can also be seen in Figure 3b, where the sensitivity surfaces have large planes at $z \approx 4$ years, and small buffers either cause a lengthy recovery or no recovery to be calculated. Secondly, missing data will affect the calculations. In the air and soil temperature, the missing data in CTE1 (Figures 1c, d) means the daily data calculates a recovered state (Figures 3c, d, Table 1) if the initial closeness buffer is large enough and there is little to no data for 6 months after the recovery day. Thirdly, abiotic data with high amounts of noise will be affected more by the summarization methods. The high daily variance in the canopy saturation data can be seen with the results in Figure 4d, with the daily data differences never even reaching 100% of the acute change for the post small-difference maximum during the experiment (the daily data sensitivity surface does not exist because Equation 3 cannot be satisfied).

Despite these shortcomings, the reported recovery times in Table 1 show evidence of being good estimates of recovery times that consider data seasonality instead of just using annual summary data estimates. The CTE annual averages showed convergence in their time series (dashed lines on Figures 1, 2) in approximately the same time as the recovery metrics (gray circles on Figures 1 and 2), but necessarily round up to the next year (or longer if the recovery is late in the year). For example, solar radiation after CTE1 recovered in a calculated 4-5 years (Table 1), or in year five, and the black and red dashed lines in Figure 1a cross at the fifth point. Throughfall after CTE1 is the most severe exception. The annual averages support recovery in 3-4 years (Figure 1b) but the calculations for recovery resulted in 9 years (Table 1).

### 3.3 Comparing Experimental Manipulations, Field Observations, and Satellite Data

The passage of hurricane María, 2.8 years after the second experiment, happened when most of the abiotic factors had not recovered and a few had just recovered. Temperatures after CTE1 and CTE2 recovered in around 2 years, almost half the time it took solar radiation to recover after CTE1, and a less than a third of the time it took throughfall to recover (Table 1). The effect of Hurricane María was smaller on the treated plots than the control plots, such that the absolute level of abiotic disturbance on the treated plots was smaller than on the control plots (Figures 1, 2). It is expected that the abiotic fluctuations from the hurricane would be smaller in the unrecovered treated plots than in the control plots since there is less vegetation to disturb. The fluctuation is smaller, but furthermore for most of the abiotic factors, the treated plots are closer to the recovered state after the hurricane than are the control plots. For example, there is more solar radiation reaching the forest floor in the treated plots than in the control plots before hurricane María, but after the hurricane there is less solar radiation reaching the forest floor in the treated plots than in the control plots (Figure 1a). The same scenario can be seen in the throughfall (Figure 1b), the temperatures to a lesser extent (Figures 1c, d), the soil moisture profile (Figure 2c), and the litter saturation (Figure 2e). The air relative humidity has the opposite scenario, showing treated plots closer to the recovery state of less humidity in the air after the hurricane (Figure 2a).

Overall, the patterns of acute changes across the abiotic factors from the experiments and the hurricane María are similar (Table 1). With only 0.09 ha plots, edge effects of the non-disturbed forest were expected to lessen the effectiveness of the

experiments in simulating hurricane disturbance; yet, the acute changes showed that CTE2 was the most immediately disruptive event across the abiotic factors, more so than the hurricane. The soil moisture increased much more in the treated plots of CTE1 and CTE2 than in the acute change calculated before to after the hurricane. However, it is impossible to know the true 'control' (no hurricane scenario) soil moisture level post-hurricane.

The satellite data have somewhat similar characteristics to the field data in the control plots (blue vs. black lines in Figures 1a, 1c, 2b) in that the magnitude of the acute change is similar (Table 1) and the responses to the summer 2015 drought and hurricane María are in the same direction. Before the hurricane, the (MODIS LAI-estimated) solar radiation satellite data look very similar to the field data, but they show a smaller change after the hurricane (Table 1) and faster recovery down to previous values than did the field data (Figure 1a). The LEF lost 51% of the initial greenness in hurricane María, but the U.S. Caribbean overall lost 31% of its initial greenness (Van Beusekom et al., 2018), so for the hurricane disturbance, including area outside the forest would be expected to dampen the measurement of the LAI hurricane disturbance signal. The (MODIS LST-estimated) temperature satellite data plot between the field air temperature data measured below the canopy and that measured above the control canopy at the tower weather station (black and green lines respectively, Figure 1c), as might be expected from a LST representative of surface energy balance. These data were strongly affected by the hurricane and quick to recover. The (AMSR2-estimated) shallow soil moisture satellite data have very large spatial smoothing (10 km resolution, containing non-forest and thin forest areas), showing a drier soil than the CTE. These data were also strongly affected by the hurricane and appear to recover quickly (Figure 2b).

**4 Discussion**

The responses in the CTE plots from after hurricane María were very similar to the responses after the two trimming events, which was the aim; nonetheless, it is encouraging how well the experiments worked. However, lacking a control plot for the actual hurricane response, the differences in the seasonal timing of the experiment treatments and hurricane María, as well as sensitivity of the calculations of actual hurricane effects to the data smoothing, make direct comparison of acute changes from the experimental events and actual hurricane disturbances challenging. For these reasons, the quantification of the acute changes in the experimental setup is most useful as a measure of the effect of a hurricane on the abiotic environment, while the quantification of the acute changes from the actual hurricane serves best as a comparison between the field and satellite data, and between the relative effects on each abiotic factor for the CTE and the hurricane.

The smaller effect of Hurricane María on the treated plots provides evidence that when frequent hurricanes happen, the forests will exhibit abiotic resilience, and thus possibly forests with an intermediate hurricane frequency will have larger abiotic fluctuations due to disturbance than forests with infrequent or frequent hurricanes. Supporting evidence has been found also in the biotic factors of the forest after hurricane María (Hogan et al., 2018), with analysis suggesting tree demographics (the rates of species and stem mortality and growth) were the most dynamic in areas which had the chance to grow some (but not

all) trees past the pioneer stage. Intermediate disturbance has long been suggested to keep systems as far from equilibrium states as possible, with the important effect of driving ecosystem diversity (Connell, 1978). Frequent disturbance in the LEF could be regarded as less than a decade (because abiotic factors have not recovered in this time frame), with intermediate frequency longer than a decade but still less than the 60-year long-term return interval for hurricane disturbance in this forest. The 60 year time frame has been estimated to be a long enough period to achieve steady state for time-length of biomass turnover (Scatena, 1995), so disturbances, for this system, might be considered infrequent if they happen less than every 60 years.

It is well-known that there are issues of scale when comparing outright the values of large-pixel satellite observations to point field observations (Wu and Li, 2009), but the faster recovery seen in the satellite data is interesting. The MODIS LAI may be measuring some low vegetation that grows back rapidly and not recovering canopy, thus decreasing the estimated satellite solar radiation back to undisturbed values more quickly than seen in field observations. The (MODIS LST-estimated) temperature satellite data and the (AMSR2-estimated) shallow soil moisture satellite data may have had large acute changes and quick recoveries because they are measuring more than just the forest. Above canopy temperatures are included in the energy balance LST data and low-permeability areas that flood and dry out are included in the AMSR2 data.

Two of the primary factors, light and water, changed dramatically after the disturbance events (Table 1, Figures 1a, 1b). Across the three events, the range of the percentage change in understory solar radiation after disturbance was quite large (214 to 919%); it is likely that a sizeable portion of the range is due to the different seasonal timing of the events. The 1998 hurricane Georges was estimated to have changed the forest light by almost 400% (Comita et al., 2009), which is within the range seen here. The response of reduced understory light and throughfall (Table 1) was found here to last much longer than the 18 months concluded previously (Richardson et al., 2010). However, it was noted in a related study (Shiels et al., 2010) that the control plot understory solar radiation appeared to be still recovering from the 1998 hurricane Georges in measurements made in 2003, 5 years after hurricane Georges. The recovery times calculated here support a continuing recovery from Georges in 2003, as the recovery of solar radiation is estimated at 4-5 years (Table 1) and the recovery of throughfall is estimated at 4 least years by the sensitivity study of the quantitative metrics (Figure 3b). This study had additional information from the second experimental trimming, as well as a longer record of analysed data from the first trimming and new methods to make a more-continuous record from the intermittent field data. The response may appear in the drier darker season as being recovered (e.g., January 2008, 3 years post-trimming), but it is clear with the longer record that the response is slower to recover. Temperatures of air and soil were much more robust in respect to the changes from the events versus their annual seasonal cycle changes, with approximately 3% air and 6% soil acute increases on average, or +0.7 °C air and +1.4 °C soil, recovered at best estimate by 2-2.5 years (Table 1, Figures 1c, 1d). But these changes may still be significant to biotic factors. Other studies show that gross primary productivity of the forest is highly sensitive to small increases in air temperature greatly increasing canopy temperature (Pau et al., 2018), so this change that is amplified in the hottest parts of the year (Figures 1c, 1d) should not be

discounted. As hurricane intensity is expected increase with climate change, there could be a compounding effect of hurricanes and global warming in the future.

420 Abiotic factors that change because of primary factor changes, or secondary factors, have more complicated recovery paths than the primary factors. Specific timelines for recovery would be expected to be highly influenced by the tree species and soil types, and the rates seen here for all abiotic factors would not necessarily apply to all hurricane-affected tropical forests. Nevertheless, general patterns might be expected to hold. All the secondary factors were clearly affected by the summer 2015 drought and subsequent long-term rainfall levels, as seen by the large magnitude decreases in summer 2015 and the recovery 425 afterwards in air relative humidity, soil moisture, and leaf saturation (Figure 2). However, daily patterns of the relative humidity in the air and leaf saturation under the canopy were significantly influenced by the temperature and light inputs (based on the results of the residual correlations), while soil moisture may not be influenced much by these inputs. The soil moisture and litter saturation responses from the first trimming present different conclusions when analysed along with the nearly continuous in situ measurements after the second trimming. Previous studies found very quick recovery of these factors, 3 months and 18 430 months, respectively (Richardson et al., 2010). However, re-analysis of the data after the first experimental trimming: separating the data into control and treated plots; calculating volume-based percentages of water in the soil and litter instead of mass-based percentages; and most significantly, looking at the trimonthly collected data from CTE1 in light of the nearly continuously collected data from CTE2, led this study to draw different conclusions of most likely longer recovery times (Table 1).

435

The soil moisture increases in all three trimming events (including the hurricane) but the magnitude of the acute change and the time till recovery appears highly dependent on the amount of rainfall (Table 1, Figure 2b, 2c). Differences between treated and control sites appear pronounced in dry periods (e.g., spring 2006 and summer 2015), with wet periods obscuring the differences in the sites when the soil may be approaching saturation (e.g., summer 2006). However, the recovery process 440 happens mostly monotonically in the smoothed time series and the soil profile may be near or at recovery at 2.8 years (Table 1). Soil moisture is higher after disturbance because there is more throughfall and less transpiration (no leaves), but once the leaf area starts to recover the soil moisture recovers quickly.

Conversely, during the dry periods the differences between treated and control sites are obscured for the leaf saturation data. 445 The litter leaves in the second trimming were measured to be wetter and drier following the trim, and not uniformly drier as concluded previously (Figures 2d, 2e). Data from the second trimming and hurricane María shows that the litter was more saturated immediately following the events, and the low canopy leaves were drier. During periods of low rainfall, the treated plots dry out faster than the control plots, in both litter and low canopy leaves. Sometimes this results in the leaf saturation being lower in the treated plots than the control plots (e.g., summer 2015 and spring 2017). When the rainfall increases after a 450 dry period, during the late-summer rainfall, the treated plot leaf saturation increases much faster than the control plots,

suggesting the long-term effect of disturbance on leaf saturation is a more dramatic modulation in saturation by rainfall. Other studies in completely different ecosystems, southeastern United States, have seen that litter is able to become more saturated after large storms than before the storms, and they attribute this to the addition of new debris being able to hold on to more water (Van Stan et al., 2017). The litter saturation data from the first experimental trimming (data from 2005-2007) do not contradict this conclusion, but due to their record length and collection interval (trimonthly) they are not overly conclusive.

The results do not support a longer or shorter recovery time interval for the second treatment, ten years after the first (Table 1). The results showed that quantifying recovery times using sub-annually summarized time series to homogenize data from several sources was a worthwhile effort, in that the abiotic factors can be sorted into quicker and slower recoveries in sub-annual lengths. However, the definition of the 'recovered point' in time will be dependent on what biotic life considers 'normal', necessarily different for every organism. The recovery times presented here for different abiotic factors are a starting point for other researchers to frame the changes found in biotic factors post-hurricane. However, the trade-offs in recovery time calculation between extracting the seasonal data signal and capturing the daily variation, and the influences of data noise and variability point to the difficulty of quantifying recovery in an environmental system.

Climate projections predict Puerto Rico air temperature will be +2 °C warmer in the coming century and rainfall will be -20 to -30% smaller in the fall and summer wet seasons (Hall et al., 2013; Karmalkar et al., 2013). Effects from future hurricanes on the abiotic factors will be on top of this background change. This means a hurricane could add an acute effect of almost 50% more to the temperature increase, with a recovery of over 2 years (Table 1). The throughfall after a hurricane was found to increase >100% with a long recovery of up to 9 years (Table 1). But, given the climate projections of more events like the summer 2015 drought, the more noteworthy effect of future hurricanes may be the litter and low canopy leaves drying out much faster in the drought and saturating faster with rain after the drought. This will create a much more dynamic environment of leaf wetness, which may have implications for biotic factors.

**5 Conclusions**

The way abiotic characteristics are disturbed and the speed at which they recover will be key to the continued existence of tropical forests under a climate with more intense hurricane activity. Climate projections predict changes that will exacerbate the effects of hurricanes of increasing temperature and dynamically changing leaf wetness. There is evidence here that intermediate hurricane frequency will have the most extreme abiotic response (with evidence on almost all abiotic factors tested) versus infrequent or frequent hurricanes, and that satellite data may show a faster recovery than field data looking at canopy response and soil moisture. Caution must be exercised when declaring the recovered point of a forest, as full abiotic canopy closing may take half a decade or longer and not all abiotic factors recover monotonically. Abiotic factor responses to hurricanes are not included in current climate projections. Results from detailed manipulative experiments such as this study are needed in order to begin to quantify abiotic factor responses to hurricanes to add to the climate projections.

## Data Availability

The CTE data are hosted on the USDA Forest Service Research Data Archive at https://doi.org/10.2737/RDS-2019-0051 (González et al., 2019).

## Author Contribution

AEVB designed and carried out the mathematical analysis. GG, JKZ, and AR designed, supervised, and carried out the experiment. GG installed sensors and oversaw data collection of solar radiation, temperature, air relative humidity, soil
moisture, and leaf saturation after CTE2. SS helped execute CTE2 and supervise the establishment. AEVB prepared the manuscript with contributions from GG and AR.

## Acknowledgements

The authors thank Carlos Estrada, Samuel Matta, Samuel Moya, María M. Rivera, Humberto Robles, and Carlos Torrens for assisting field data, and Ariel E. Lugo and Michael Richardson for comments on an earlier version of the manuscript. This
research was funded by the Luquillo Critical Zone Observatory (National Science Foundation grant EAR-1331841) and the Luquillo Long-Term Ecological Research Site (National Science Foundation grant DEB-1239764). All research at the International Institute of Tropical Forestry is done in collaboration with the University of Puerto Rico. Any use of trade, product, or firms' names is for descriptive purposes only and does not imply endorsement by the U.S. Government.

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

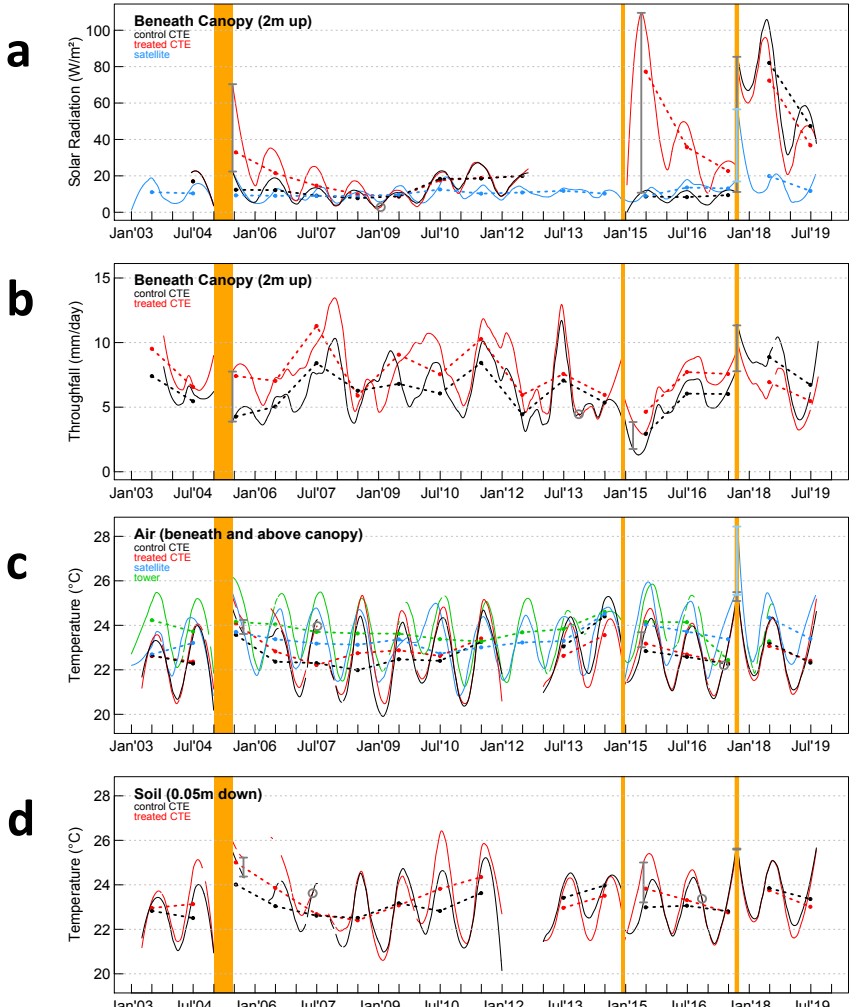

**Figure 1: Primary factor time series, or factors that change due to the initial disturbance changes.** Orange vertical lines are the periods of canopy trimming experiment (CTE) 1, CTE2, and hurricanes Irma and María (appear as one line), sequentially. Daily values of data are represented with fitted smoothed lines and thick dotted dashed lines connect annual averages of daily data to aid in visualization of the differences between the time series. Red lines are from treated areas and black lines are from control areas (until the hurricanes) beneath the canopy. Green lines are from the tower weather station above the CTE control canopy and blue lines are from satellite data. Vertical bars show the acute change after an event for CTE data (gray) and satellite data (blue). The time of recovery from each experiment calculated off the smooth data (if seen) is marked with a gray circle, Plots show a) solar radiation beneath the canopy; b) throughfall; c) air temperature; and d) soil temperature.

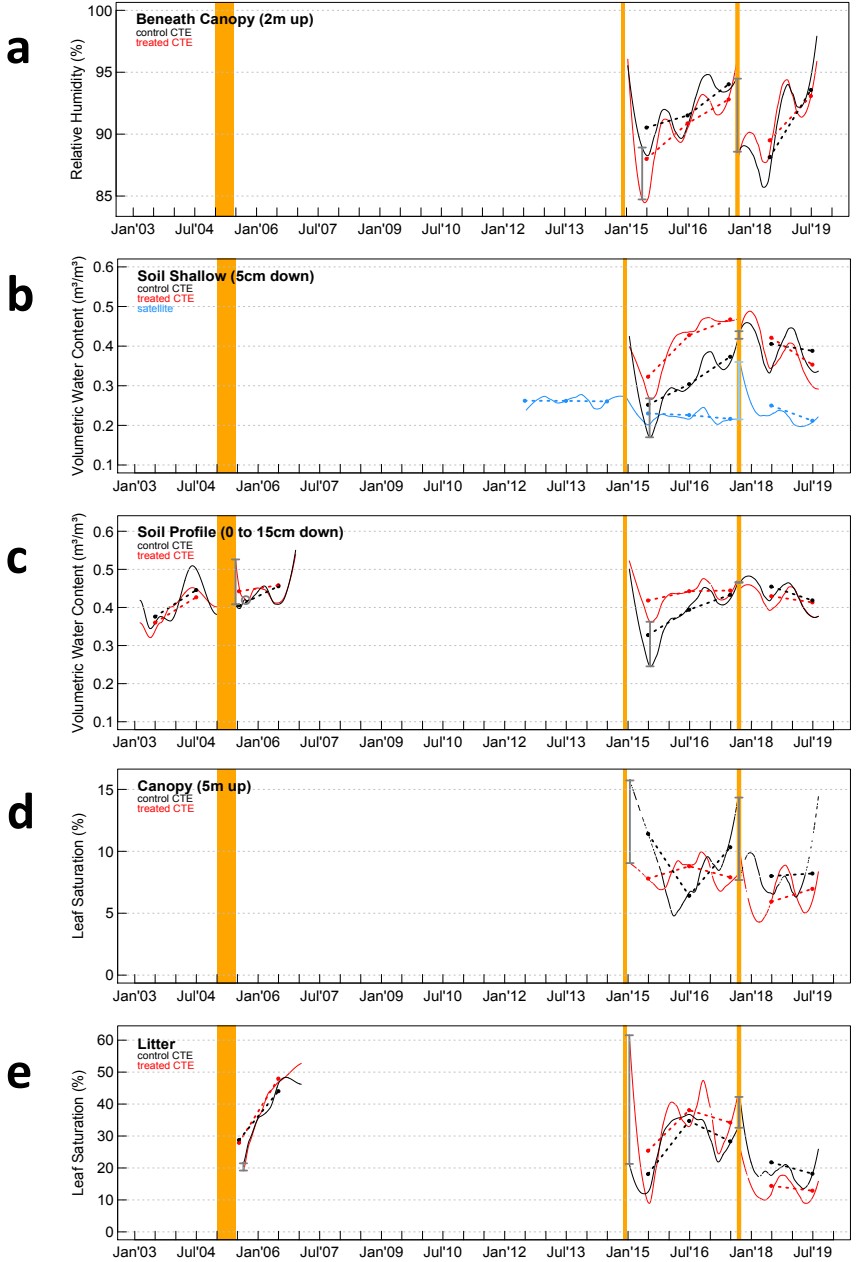

**Figure 2: Secondary factor time series, or factors that change because of primary factor changes. All markings are the same as in Figure 1. Plots show a) air relative humidity; b) soil moisture shallow; c) soil moisture profile; d) low canopy leaf saturation; and e) litter leaf saturation.**

| | Change from CTE | | | | Change from Hurricane | | | | Recovery Time | | | | | |
| --- | --- | --- | --- | --- | --- | --- | --- | --- | --- | --- | --- | --- | --- | --- |
| | CTE1* | | CTE2* | | Instruments† | | Satellite† | | CTE1‡ | *sm-mn* | *daily* | CTE2‡ | *sm-mn* | *daily* |
| **Solar Radiation** | 214 % | *48 W/m²* | 919 % | *99 W/m²* | 666 % | *74 W/m²* | 234 % | *40 W/m²* | 4.1 yrs | *4.0* | *5.1* | >2.8 yrs | *>2.8* | *>2.8* |
| **Throughfall** | 100 % | *4 mm/day* | 119 % | *2 mm/day* | 46 % | *4 mm/day* | | | 8.9 yrs | *>9.9* | *>9.9* | >2.8 yrs | *2.5* | *2.8* |
| **Temperature Air** | 3 % | *0.6 °C* | 3 % | *0.7 °C* | 2 % | *0.4 °C* | 12 % | *3.1 °C* | 2.5 yrs | *>9.9* | *>9.9* | 2.5 yrs | *1.2* | *2.2* |
| **Temperature Soil** | 3 % | *0.9 °C* | 8 % | *1.8 °C* | 0 % | *0.05 °C* | | | 2.4 yrs | *8.1* | *>9.9* | 2.0 yrs | *2.8* | *>2.8* |
| **Relative Humidity** | | | -5 % | *-4 %* | -6 % | *-6 %* | | | | | | >2.8 yrs | *>2.8* | *>2.8* |
| **Shallow Soil VWC** | | | 58 % | *0.1 m³/m³* | 5 % | *0.02 m³/m³* | 67 % | *0.1 m³/m³* | | | | >2.8 yrs | *>2.8* | *>2.8* |
| **Soil Profile VWC** | 29 % | *0.1 m³/m³* | 48 % | *0.1 m³/m³* | 0 % | *-0.002 m³/m³* | | | 0.7 yrs | *1.2* | *1.2* | >2.8 yrs | *>2.8* | *2.8* |
| **Saturation Canopy** | | | -42 % | *-7 %* | -46 % | *-7 %* | | | | | | >2.8 yrs | *>2.8* | *>2.8* |
| **Saturation Litter** | -11% | *-2 %* | 189 % | *-40 %* | 30 % | *-10 %* | | | >2.1 yrs | *1.0* | *1.5* | >2.8 yrs | *>2.8* | *>2.8* |

* First column is percentage change from control; second column in italics is absolute change from control

† First column is percentage change from before hurricane María; second column italics is absolute change from before hurricane María

‡ First column is recovery time based on smooth data; second and third columns are based on smooth-mean (sm-mn) and daily data, respectively, with units in years

**Table 1: Change after each disturbance event as seen by field instruments and satellites, and recovery time from the canopy trimming experiments (CTE1 and 2).**

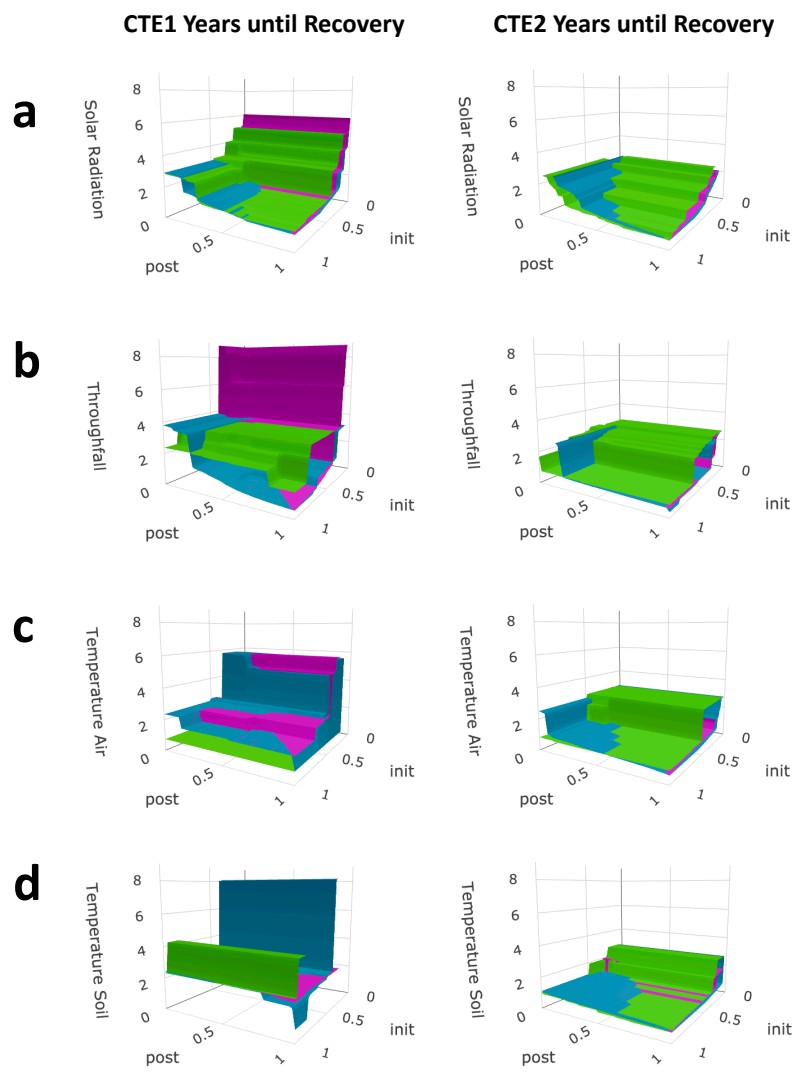

**Surface colors:** Smooth-mean data   Mean data   Daily data

625

**Figure 3: Sensitivity studies on primary factor calculated recovery times. Calculated years until recovery for each factor are plotted on the vertical axis, with variables of buffers on the other axes. The buffers are the required fraction of the acute change the difference in control and treated plots must approach, on the recovery day for initial closeness (init) and for 6 months after for post small-differences (post). Plot column one is for after canopy experiment (CTE) 1 and column 2 is for after CTE2. Surface color represents data summary methods as indicated, and absence of values for specified buffer conditions means the factor did not recover during the time period of the experiment with the specified buffers. Plots show a) solar radiation beneath the canopy; b) throughfall; c) air temperature; and d) soil temperature.**

**CTE1 Years until Recovery**          **CTE2 Years until Recovery**

a

b

c

d

e

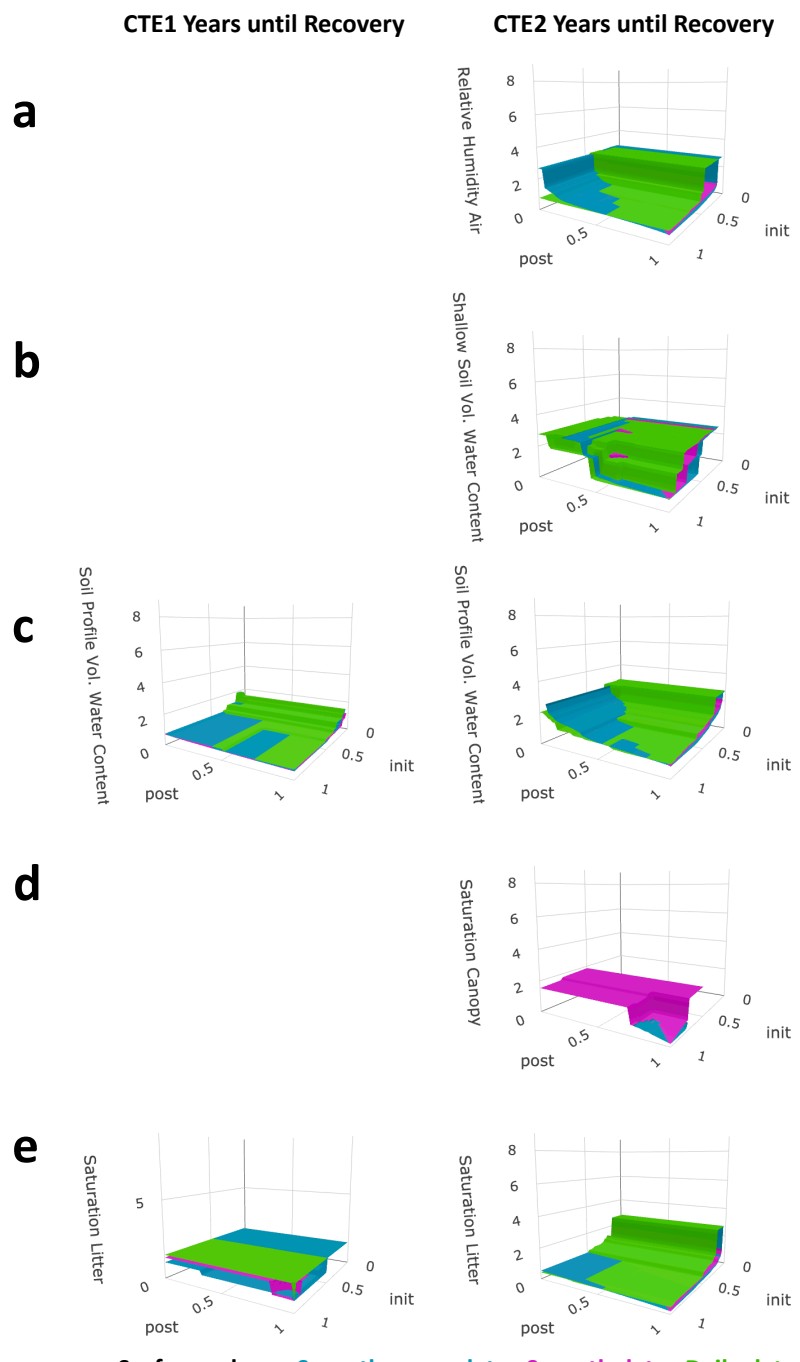

**Surface colors:** Smooth-mean data  Smooth data  Daily data

**Figure 4: Sensitivity studies on secondary factor calculated recovery times. All markings are the same as in Figure 3. Plots show a) air relative humidity; b) soil moisture shallow; c) soil moisture profile; d) low canopy leaf saturation; and e) litter leaf saturation.**