# Peer review of "Understanding Tropical Forest Abiotic Response to Hurricanes using Experimental Manipulations, Field Observations, and Satellite Data"

_Biogeosciences, 2019_

## Referee Comment (RC1) · Anonymous Referee #1 · 1 Nov 2019

Van Beusekom et al. present measurements of the forest abiotic environment following experimental and natural disturbances in the Luquillo forest in Costa Rica over a period of 16 years. They use this information to assess the recovery time of different variables. Measurements such as these can provide valuable insights into the mechanisms which govern a particular ecosystem response – particularly when combined with measurements or modelling of plant responses. The paper is clearly written and presented, and the measurements are well-described and, as best as I can judge, appropriately controlled for changes in measurement technique. However, the key to the

story of the paper is the definition of recovery time, and this appears to be somewhat arbitrarily defined with significant consequences for the results. On this basis, I cannot recommend the paper for publication in its current form.

Recovery time is defined in the paper as the point when the treated data timeseries crosses the control data timeseries and afterwards stays within 15

The choice of x and y is also critical, however. x=15

Even if one just eyeballs the plots, whilst one can be fairly confident about recovery for solar radiation, for throughfall it is much less clear (there is even divergence in 2014 following the supposed point of recovery, making it questionable whether recovery had even occurred). The definition of recovery time therefore needs some careful thought and sensitivity testing to give confident that the results are robust to the method used.

**Minor comments**

Line 94. Were Campbell sensors used after 2015 as well? In the previous paragraph it indicates not, but here that they were.

L187. Is this really resilience? There is presumably just less vegetation to be disturbed, which naturally leads to a smaller fluctuation. I would argue it just leads to lower amplitude of variability.

L188. "greater disturbance" is not clear. Perhaps, "greater fluctuations in the measured abiotic variables due to disturbance"?

L190. What exactly does it mean that "tree demographics were... dynamic"? Does this refer to the mix of ages in the forest, the rate of growth, the rate of turnover?

---

## Referee Comment (RC2) · Anonymous Referee #2 · 13 Nov 2019

General Comment

This study integrated the observations both from in-situ and satellite platform for studying the dynamics of vegetation change in Luquillo Experimental Forest. Two canopy trimming experiments, one in 2004 and another in 2015, were designed as control experiments to reveal the vegetation recovery in response to the wind damage to the trees, especially for the case caused by the tropical storms (Irma and Maria) in 2017. The authors reported long term and continues time series of under-canopy solar radiation, throughfall, air temperature (under and above), soil water, and relative humidity

and leaf saturation in the manuscript. This work can provide an insight into the vegetation recovery due to the wind disturbance in the tropical climate zone. However, the structure of the manuscript and approach for analysis the data are a bit confusing. I suggested that the authors provide a general review of the vegetation recovery in the introduction section and try to focus on the study results for the tropics. Here, I provided a few studies (listed in the reference) including observation and modeling works which are relevant for providing a general review of the wind disturbance research. The introduction of the canopy trimming experiment can move to the methodology section which can be the design of canopy trimming and natural disturbance events. Along with this discussion, the method applied for this study to identify the recovery period is questionable, and the authors didn't include or calculate the uncertainty caused by instruments, sampling approaches, or data analysis (smoothing). Regarding the issue for identifying the recovery period, I recommended the authors to analysis the annual maximum observations, for example the study made by Lin et al. (2016). By comparing annual maximum values over a long-term time series is helpful to identify the status of vegetation recovery period. I had several specific comments for the authors to improve the current version of this manuscript.

Specific comments

1.Using the measurement of wetness of litter leaves and soil water to understand the canopy recovery physically is not reasonable. Although the wetness of litter leaves and soil moisture can be affected by the coverage of the over-story canopy, the magnitude of soil moisture and litter leaves are fixed which might only depend only on the soil property and leaf types. Please explain how to use the observation of soil moisture and wetness of litter leaves to reveal the status vegetation recovery.

2.P2L61: (wetness of canopy and litter leaves) How to determine the wetness of canopy leave and litter leaves.

3.P3L79: "locally to the points", Can you show the original points in your results?

4.P3L90: The MODIS only measured the sink temperature of the surface. Why did the authors compare the air temperature observations to the MODIS LST observations?

5.P3L92: How many 5TM sensors were deployed for soil water observation? What is the minimum requirement for avoiding the spatial heterogeneity under canopy at this study site?

6.P4L115-L124: Too many details were lost or cannot be found. For example, the relationship between the 8-day MODIS LAI and 8-day in-situ solar radiation was built up for converting the MODIS LAI to solar radiation for the study site, but the authors didn't present this information and uncertainty.

7.P5L149-150: The reason for applying 1year smooth window is not clear, please explain in the method section.

8.P5L159-161: The way for justifying the recovery period is not clear, please explain the method in detail.

9.P6L169-176: I didn't understand why the authors reported the residuals of the statistical analysis in this paragraph. Is this information helpful for understanding the uncertainty of various measurements?

10.In the Discussion section: It is very difficult for me to find/justify the information of the recovery periods, such 10 years, 2.8 years and others values from Figs 1 and 2. I recommended the authors to indicate such a piece of information both in this section and key Figs.

References Chen et al., 2018: Simulating damage for wind storms in the land surface model ORCHIDEE-CAN (revision 4262), Geosci. Model Dev., 11, 771-791. Lin et al., 2017: Impacts of increasing typhoons on the structure and function of a subtropical forest: reflections of a changing climate, Scientific Reports, 4911. Mitchell S.J., 2013: Wind as a natural disturbance agent in forests: a synthesis, Forestry, 86, 147-157. Negrón-Juárez et al, 2014: Multi-scale sensitivity of Landsat and MODIS to

forest disturbance associated with tropical cyclones, Remote Sensing of Environment, 140, 679-689. Rebane et al., 2019: Direct measurements of carbon exchange at forest disturbance sites: a review of results with the eddy covariance method, Scandinavian Journal of Forest Research, https://doi.org/10.1080/02827581.2019.1659849 Uriarte, et al, 2019: Hurricane María tripled stem breaks and doubled tree mortality relative to other major storms, Nature Communication, 1362. Wu et al., 2019: Sensitivity analysis of the typhoon disturbance effect on forest dynamics and carbon balance in the future in a cool-temperate forest in northern Japan by using SEIB-DGVM, Forest Ecology and Management, 451, 1, 117529.

---

## Author Response (AR1)

[revised manuscript text omitted]

RC1: Van Beusekom et al. present measurements of the forest abiotic environment following experimental and natural disturbances in the Luquillo forest in Costa Rica [sic, should be Puerto Rico] over a period of 16 years. They use this information to assess the recovery time of different variables. Measurements such as these can provide valuable insights into the mechanisms which govern a particular ecosystem response – particularly when combined with measurements or modelling of plant responses. The paper is clearly written and presented, and the measurements are well-described and, as best as I can judge, appropriately controlled for changes in measurement technique. However, the key to the story of the paper is the definition of recovery time, and this appears to be somewhat arbitrarily defined with significant consequences for the results. On this basis, I cannot recommend the paper for publication in its current form.

Recovery time is defined in the paper as the point when the treated data timeseries crosses the control data timeseries and afterwards stays within 15

The choice of x and y is also critical, however. x=15

Even if one just eyeballs the plots, whilst one can be fairly confident about recovery for solar radiation, for throughfall it is much less clear (there is even divergence in 2014 following the supposed point of recovery, making it questionable whether recovery had even occurred). The definition of recovery time therefore needs some careful thought and sensitivity testing to give confident that the results are robust to the method used.

AUTHORS: We have expanded the reasoning on the recovery methods and added sections on sensitivity testing. In the Introduction, we agree that recovery is an arbitrarily defined point, saying "This study attempts to quantify abiotic response as acute changes from a hurricane disturbance (experimental or otherwise) and recovery from the changes, for primary and secondary factors. Quantifying the responses makes it possible to assess if the experimental trimming data and satellite data are reasonable sources for studying the effect of hurricane disturbance and appear to be measuring the same abiotic system, as well as appreciate if different events cause substantially different responses. This study does not attempt to determine what amount of recovery is considered 'normal' conditions to biotic life, or in other words what would affect tertiary factors, but instead quantifies changes in the abiotic factors that can be used to frame the changes found in biotic factors post-hurricane in many previous studies including those of biotic abundance (Shiels et al., 2015), soil biochemistry (Arroyo and Silver, 2018), and plant reproduction (Zimmerman et al., 2018)."

We added two paragraphs in the beginning of the Methods to explain the reasoning for the recovery metric. "The LOESS degree of smoothing is contingent on the size of the local neighborhood, which here was always chosen to be one year of data around each point. The yearly smoothing was done to extract the larger signal from the data and to homogenize the different collection intervals of the data. The automated sensor field data captured larger amounts of background noise than the temporally smoothed rain funnel data and the geographically smoothed satellite data; and to a lesser extent, the geographically smoothed soil sample, litterbag, and canopy photo data. The one-year smoothing neighborhood was chosen to be longer than the longest length of time between repeat measurements across all data types and methods.

Calculations for abiotic responses were made on the resulting time series with the one-year smoothing. Recovery after a CTE experiment was defined as the point in time that the treated data time series crosses the time series of the control data, afterwards which the difference between the treated and control data stays within a 15% buffer of the control data for a year, or until the next event. This could be a conservative measure for biotic recognition, but from an abiotic point of view the 15% buffer corresponds with visual recovery in the time series. Other studies have defined recovery as the year in which the annual maximum value (of the disturbed area) returns to a previous annual maximum value (assumed representative of undisturbed conditions; Lin et al., 2017). While the method used here is dependent on the size of the smoothing neighborhood; it is able to make use of the parallelly collected control data to calculate more precise recovery lengths than a year. Furthermore, in a frequently disturbed regime such as the LEF, it is difficult to say what year would be representative of undisturbed conditions. Time series were also analyzed to calculate acute change from disturbance. The acute change after the hurricane was defined as the change in the control time series or the satellite time series from right before the hurricane to right after the hurricane, September 20, 2017. The acute change after an experiment disturbance event was defined as the maximum difference between the treated and control time series (in relation to the control time series) on any day between the last day of the canopy trimming (spring 2005, December 2014) and of the next September 20 (year 2005 and 2015, respectively), so that the experimental changes could be compared to the hurricane changes. Sensitivity tests were performed to see how the calculated recovery lengths differed with smaller and larger buffers than the 15 %, as well as how the recovery lengths and acute changes differed with smaller and larger smoothing neighborhoods than the one year."

We have also now reported the results of the sensitivity tests on smoothing neighborhood size and the 15% buffer, as suggested. The recovery times and disturbance changes reported in the Table results are for the most part robust against smoothing amounts. In the results, it says "Sensitivity tests were performed using different LOESS smoothing neighborhoods and altering the size of the 15% buffer. The calculated recovery times are very robust to altering in the size of the neighborhood from half as large to twice as large (neighborhoods of 0.5-2 years), with a mean of less than ±0.2 years for any neighborhood size. Larger neighborhoods than the one-year reported in Table 1 disproportionally effect the calculated recovery times of the coarser data, throughfall and CTE1 litter saturation (Figures 1b, 2e). Smaller neighborhoods than the one-year reported in Table 1 disproportionally affect the calculated recovery times of the noisier data and the data with many missing observations, throughfall and CTE1 air and soil temperatures, respectively (Figures 1b-d). Allowing the buffer (inside which the control and treated plots are said to be similar enough to warrant a recovered state) to be from half as large to twice as large (buffers of 7.5-30%) only affects the calculation of the recovery lengths of CTE1 solar radiation, CTE1 throughfall, and CTE1 and CTE2 litter saturation (Figures 1a, 1b, 2e and Table 1). Solar radiation is calculated to recover after CTE1 somewhat quicker, in 5 years (from 6 years) with a larger buffer, and it does not recover in the 7.6 years if the buffer is shrunk to 7.5%. Throughfall recovery calculation does not change if the buffer is larger, but it does not recover in the 9.9 years of CTE1 if the buffer is smaller. Litter saturation is calculated to recover after CTE2 in 0.7 years right after the summer 2015 drought (down from >2.8 years) with a larger buffer (and still in 1.0 years after CTE1, and it does not recover in the 2.1 years after CTE1 if the buffer is shrunk to 7.5% (and still not in the 2.8 years after CTE2). The calculated changes after an experimental disturbance event are fairly robust to altering the size of the neighborhood (absolute changes are on average less than ±15% different), but the calculated changes after the hurricane can be quite affected if the neighborhood is expanded, making the time series smoother at the end points before and after the hurricane (Figures 1, 2)."

We also added some discussion around this, saying at the beginning of the Discussion: "However, the differences in the seasonal timing of the experiment treatments and hurricane María, as well as sensitivity of the calculations of actual hurricane effects to the data smoothing, make direct comparison of acute changes from the experimental events and actual hurricane disturbances challenging. The quantification of the acute changes in the experimental setup is useful as a measure of the effect of a hurricane on the abiotic environment, while the quantification of the acute changes from the actual hurricane serves best as a comparison between the field and satellite data, and between the relative effects on each abiotic factor for the CTE and the hurricane."

680 At the end of the Discussion, we added "The results do not support a longer or shorter recovery time interval for the second treatment, ten years after the first (Table 1). The results in the sensitivity tests showed that quantifying recovery times using smoothed time series to homogenize data from several sources was a worthwhile effort, in that the abiotic factors can be sorted into quicker and slower recoveries, with results robust to the smoothing method. However, the definition of the 'recovered point' in time will be dependent on what biotic life considers 'normal',

685 necessarily different for every organism. Across all abiotic factors, this study used a uniform buffer metric of 'within 15% agreement between control and treated plots' once the experimental response is finished, in order to quantify the length of abiotic recovery as a starting point to for other researchers to frame the changes found in biotic factors post-hurricane. The percentage of the buffer did not matter to the results in most cases, as shown in the sensitivity tests. However, the percentage did alter the results on factors with more complicated recovery paths, such as litter saturation (Figure 2e), and factors with more data variance, such as solar radiation and throughfall

690 (Figures 1a, b). This points to the difficulty of quantifying recovery in an environmental system."

RC1: Minor comments:

695 Line 94. Were Campbell sensors used after 2015 as well? In the previous paragraph it indicates not, but here that they were.

AUTHORS: Campbell temperature 107 sensors were only used before 2015, and after 2015, soil VWC was measured by CS616 sensors, which are also made by Campbell. We changed the wording in this paragraph to refer

700 to the sensors as '107 sensors' instead of Campbell sensors to avoid this confusion.

RC1: L187. Is this really resilience? There is presumably just less vegetation to be disturbed, which naturally leads to a smaller fluctuation. I would argue it just leads to lower amplitude of variability.

705

AUTHORS: We clarified this statement, pointing out that the treated plots are closer to recovery after the hurricane than the control plots are. "The passage of hurricane María, 2.8 years after the second experiment, showed a smaller effect on the treated plots than the control plots, such that the absolute level of abiotic disturbance on the treated plots was smaller than on the control plots (Figures 1, 2). It is expected that the abiotic fluctuations from the

710 hurricane would be smaller in the unrecovered treated plots than in the control plots since there is less vegetation

to disturb. The fluctuation is smaller, but furthermore for most of the abiotic factors, the treated plots are closer to the recovered state after the hurricane than are the control plots. For example, there is more solar radiation reaching the forest floor in the treated plots than in the control plots before hurricane María, but after the hurricane there is less solar radiation reaching the forest floor in the treated plots than in the control plots (Figure 1a). The same scenario can be seen in the throughfall (Figure 1b), the temperatures to a lesser extent (Figures 1c,d), the soil moisture profile (Figure 2c), and the litter saturation (Figure 2e). The air relative humidity has the opposite scenario, showing treated plots closer to the recovery state of less humidity after the hurricane (Figure 2a).

RC1: L188. "greater disturbance" is not clear. Perhaps, "greater fluctuations in the measured abiotic variables due to disturbance"?

AUTHORS: Changed to "larger abiotic fluctuations due to disturbance".

RC1: L190. What exactly does it mean that "tree demographics were . . . dynamic"? Does this refer to the mix of ages in the forest, the rate of growth, the rate of turnover?

AUTHORS: All of the above. We added an explanation "(the rates of species and stem mortality and growth)"

RC2: General Comment

This study integrated the observations both from in-situ and satellite platform for studying the dynamics of vegetation change in Luquillo Experimental Forest. Two canopy trimming experiments, one in 2004 and another in 2015, were designed as control experiments to reveal the vegetation recovery in response to the wind damage to the trees, especially for the case caused by the tropical storms (Irma and Maria) in 2017. The authors reported long term and continues time series of under-canopy solar radiation, throughfall, air temperature (under and above), soil water, and relative humidity and leaf saturation in the manuscript. This work can provide an insight into the vegetation recovery due to the wind disturbance in the tropical climate zone. However, the structure of the manuscript and approach for analysis the data are a bit confusing. I suggested that the authors provide a general review of the vegetation recovery in the

introduction section and try to focus on the study results for the tropics. Here, I provided a few studies (listed in the reference) including observation and modeling works which are relevant for providing a general review of the wind disturbance research. The introduction of the canopy trimming experiment can move to the methodology section which can be the design of canopy trimming and natural disturbance events. Along with this discussion, the method applied for this study to identify the recovery period is questionable, and the authors didn't include or calculate the uncertainty caused by instruments, sampling approaches, or data analysis (smoothing). Regarding the issue for identifying the recovery period, I recommended the authors to analysis the annual maximum observations, for example the study made by Lin et al. (2016). By comparing annual maximum values over a long-term time series is helpful to identify the status of vegetation recovery period. I had several specific comments for the authors to improve the current version of this manuscript.

AUTHORS: Thank you for your detailed comments. We address those below. We moved the description of the experiment to the methods and we have also added the Mitchell (2013) reference in the introduction. We have greatly expanded the methodology description of the recovery metrics (with added results), citing Lin et al. (2016) as discussed under the comment 8. The papers on windthrow modeling and tree mortality do not seem to be on topic as we are concerned with the abiotic environment and not the geographical extent of the disturbance.

RC2: Specific comments

1.Using the measurement of wetness of litter leaves and soil water to understand the canopy recovery physically is not reasonable. Although the wetness of litter leaves and soil moisture can be affected by the coverage of the overstory canopy, the magnitude of soil moisture and litter leaves are fixed which might only depend only on the soil property and leaf types. Please explain how to use the observation of soil moisture and wetness of litter leaves to reveal the status vegetation recovery.

AUTHORS: We are not attempting to understand canopy recovery, but instead how the forest abiotic environment responds to the vegetation recovery, and when the abiotic environment is recovered to its pre-hurricane state. We added this sentence in the introduction "More than understanding when vegetation has recovered, it is important to understand how the abiotic environment affected by the vegetation changes recovers from the disturbance." We agree that the timeline of soil and litter moisture recovery very much depends on the types of soil and leaves involved. We are focused on the response, not the specific timeline. This comment has been added to the discussion in the section talking about the soil and litter patterns: "Specific timelines for recovery would be expected to be highly influenced by the tree species and soil types, and the rates seen here for all abiotic factors would not necessarily apply to all hurricane-effected tropical forests. Nevertheless, general patterns might be expected to hold." We have pointed out that similar patterns to the litter saturation response patterns seen here were also presented in Southeastern United States.

RC2: 2.P2L61: (wetness of canopy and litter leaves) How to determine the wetness of canopy leave and litter leaves.

AUTHORS: This is discussed in the methods, "Leaf saturation data were collected after 2015 by Decagon Devices dielectric leaf wetness sensors in the canopy leaves 5 m up from the ground and in the litter leaf layer."

RC2: 3.P3L79: "locally to the points", Can you show the original points in your results?

AUTHORS: We have added the points to the plots.

RC2: 4.P3L90: The MODIS only measured the sink temperature of the surface. Why did the authors compare the air temperature observations to the MODIS LST observations?

795    AUTHORS: We compared to see if any of the forest cover change seen in the field observations of temperature (above and below canopy) were comparable to the LST. We clarified this in the methods, saying "MODIS LST measures energy balance at the land surface, so is not representative of air temperature under the canopy but it will be affected by changes in air temperature. Annual maximums of LST and air temperature are highly correlated across the globe with correlation strongest in forested areas (Mildrexler et al., 2011), and LST has been shown to

800    respond to forest cover changes in other areas of the tropics (van Leeuwen et al., 2011)." Again in the discussion, we clarified "The (MODIS LST-estimated) temperature satellite data plot between the field air temperature data measured below the canopy and that measured above the canopy at 30 m (black and gray lines respectively, Figure 1c), giving evidence that the satellite measurements were affected by a vertically averaged Earth, as might be expected from a LST representative of surface energy balance."

805

RC2: 5.P3L92: How many 5TM sensors were deployed for soil water observation? What is the minimum requirement for avoiding the spatial heterogeneity under canopy at this study site?

810    AUTHORS: We use several sensors in each of the 6 plots to avoid this problem. We added the number of sensors to each paragraph in the methods, for each type of sensors. At the beginning of the methods, we now explain "To account for spatial heterogeneity under the canopy, multiple sensors were used in each plot were used and the results were averaged in all control and treated plots (with quality control)."

815

RC2: 6.P4L115-L124: Too many details were lost or cannot be found. For example, the relationship between the 8-day MODIS LAI and 8-day in-situ solar radiation was built up for converting the MODIS LAI to solar radiation for the study site, but the authors didn't present this information and uncertainty.

820    AUTHORS: We clarified this section by expanding description to "The Beer-Lambert law (Monsi, 1953) was used to convert the LAI data into solar radiation estimates. Annual patterns of photosynthetically active radiation (PAR) extinction coefficients needed for the Beer-Lambert law were calculated by solving the Beer-Lambert Law for the coefficients with the field-measured control plot solar radiation, and the field-measured above canopy solar

radiation, and the MODIS LAI data. The coefficients were solved for using with all data interpolated or averaged
to daily values, and only using the two years of data before the hurricane (so excluding the 2015 drought)."

RC2: 7.P5L149-150: The reason for applying 1year smooth window is not clear, please explain in the method
section.

AUTHORS: We added an expanded explanation to the start of the methods: "The LOESS degree of smoothing is
contingent on the size of the local neighborhood, which here was always chosen to be one year of data around each
point. The yearly smoothing was done to extract the larger signal from the data and to homogenize the different
collection intervals of the data. The automated sensor field data captured larger amounts of background noise than
the temporally smoothed rain funnel data and the geographically smoothed satellite data; and to a lesser extent, the
geographically-smoothed soil sample, litterbag, and canopy photo data. The one-year smoothing neighborhood was
chosen to be longer than the longest length of time between repeat measurements across all data types and
methods."

RC2: 8.P5L159-161: The way for justifying the recovery period is not clear, please explain the method in detail.

AUTHORS: We have moved this section to the methods and expanded to explain the reasoning and make it clear
that we ran sensitivity tests to find this method acceptable. We now say in the methods "
[revised manuscript text omitted]

RC2: 9.P6L169-176: I didn't understand why the authors reported the residuals of the statistical analysis in this paragraph. Is this information helpful for understanding the uncertainty of various measurements?

AUTHORS: This is the method for correlating time series: we remove seasonality and trends and correlate what is left over. We had called these 'leftovers' as 'residuals. To make this clearer, we changed the words "residuals of [data]" to "the prewhitened [data]".

RC2: 10.In the Discussion section: It is very difficult for me to find/justify the information of the recovery periods, such 10 years, 2.8 years and others values from Figs 1 and 2. I recommended the authors to indicate such a piece of information both in this section and key Figs.

AUTHORS: We added a few more references to Table 1, where all the specific recovery information is at. We also added green points on Figures 1 and 2 at the point where recovery is calculated.

---

## Author Response (AR2)

[revised manuscript text omitted]

EDITOR: Thank you for submitting a revised version of your manuscript. The manuscript has now been seen by an associate editor who despite the positive assessment of the changes made following the open discussion, recommends major revisions before sending the manuscript to the initial reviewers.

In "Understanding Tropical Forest Abiotic Response to Hurricanes using Experimental Manipulations, Field Observations, and Satellite Data" the authors present a 16 year-long time series of experimental data to describe the change and recovery of abiotic variables following a hurricane in tropical forest. Although the authors made an effort to address the comments of the reviewers, the poor structure of the manuscript makes it difficult to evaluate its scientific merits. I, therefore, propose that the authors restructure the manuscript before it is being send out for review by the initial reviewers. The revision should not simply try to please the reviewer by ticking the boxes listed below but should consist of a thorough effort guided by the referee's comment to enhance the presentation and the flow of the study.

A paragraph should be between 10 and 15 lines long and should contain a single idea. Paragraphs help the readers to absorb the information in a natural way. The manuscript contains too many paragraphs that exceeds the recommended length as well as paragraphs that touch on more than one idea/issue (L115-133). Use paragraphs correctly to increase the flow of the text.

The method section should contain a description of all methods used in the study. Over 20% of the result section should be moved into the methods section. Use subsections to better structure the methods. One way could be to have a subsection describing the experimental treatments followed by an individual subsection for each variable. At present it is written that the canopy was trimmed and readers are referred to other papers. This manuscript should be a stand-alone piece and readers should be able to grasp the essence of the trimming experiment without having to access other papers. For each variable the methods before and after 2015 should be described as well as the methods used to homogenize the time series. The section could be completed with a subsection explaining how recovery was defined and calculated. The 2015 drought plays a central role in the discussion and its length, intensity and impact should thus be characterized. The tower is mentioned for the first time in the discussion.

AUTHORS: Thank you for your comments. We have significantly restricted the manuscript following your comments, as can be seen in the tracked changes manuscript. We have rewritten the methods and results sections, following the advice here. The methods section now has subsections as suggested, and the tower data is brought up here. Each abiotic variable type has its own subsection explaining its collection and homogenization. A new subsection contains the description of the response methods. More details of the trimming are now in the introduction so the paper can stand-alone. The drought is characterized in the introduction as suggested. The previous responses to the reviewers have been edited to reflect the new changes. There are now two sets of changes to the tracked changes manuscript: the original edits in response to the reviewers' comments are under "Ashley Van Beusekom" and the changes after that (in response to the editors comments) are under "Ashley Van Beusekom2".

EDITOR: In the result sections the authors should draw the attention of the reader to the most interesting results (which should be put into context in the discussion). The current result section contains too many information on the methods or simply repeats the information contained in the captions of the figures (for example 212-215, 222-228, 249-252, 290-325, 344-370, 372-382). A major effort is needed to enhance the quality of the methods section.

AUTHORS: We completely revised the results section as suggested, removing the caption information and moving the methods to the methods section. The interesting results are now highlighted here (instead of first being brought up in the discussion as they were previously).

EDITOR: The figures basically show the time series but fail to convince at a glance what is being written about the quality of the experiments compared to effect of a real hurricane and subsequent recovery. The effect of the experimental treatments and the hurricane on the abiotic environment is being dwarfed by its seasonal variation. Statistics, for example effect sizes, could be used to focus on the figures on the effects rather than the seasonal variation. More processed figures would contribute to the potential impact of the study. Simply showing time series require the readers to do a lot of the work themselves. Can you think of better formats to show the impact and recovery in a single figure? For example, calculate the intercept and slope between control and treatment within a moving window. Plot a time series of the intercept and/or slope (with their uncertainty interval). Differences right after the disturbance should be the largest. Similarities between time series should increase when approaching recovery. An important challenge in science nowadays is to find compelling ways to present large data sets and especially the lessons that can be learned from these data sets. The present figures are not very helpful from that point of view.

AUTHORS: We removed the raw point data that one reviewer requested, and instead added lines connecting annual averages to "aid in visualization of the differences between the timeseries." The acute change and the point of recovery as indicated by the recovery metrics is marked on each figure. The methods for the figure are now discussed in the methods section, making it more clear that these figures are showing processed results and not raw data.

EDITOR: The first paragraph of the discussion is interesting but it is not backed up by the results.

AUTHORS: The acute change metrics show that CTE2 was more effective than the hurricane at disturbing the forest at the field site. We moved this part into the results to clarify the paragraph. We put more cautionary words in throughout the paragraph to emphasize that this is the start of the discussion, and that there are caveats in direct comparison of experimental and actual hurricane results.

EDITOR: L291 to 300 as well as L328-334 should be moved to the results.

AUTHORS: This has been done.

EDITOR: Specific comments
L119 recovery instead of recognition?

AUTHORS: Changed this to "It is possible that this is a conservative measure for certain biotic species' perception
of abiotic disturbance, but from an abiotic point of view the 15% buffer corresponds with visual recovery in the time series."

EDITOR: L126- seems to belong to the previous paragraph.

AUTHORS: Added a new topic sentence and tied this paragraph together better. This paragraph is now discussing the metrics we use to quantify the response, so recovery and acute change. The topic sentence is "To explore the differences between responses of different abiotic variables, a smooth time series of each variable was computed, and recovery and acute change of each variable was quantified by pre-defined metrics." Then the "recovery and acute change pre-defined metric definitions" are discussed.

EDITOR: L137 Not clear how measurements were made above the canopy. I guess a tower was used but this nowhere stated. It is mentioned in the discussion which is too late.

AUTHORS: The tower is now introduced at the beginning of the methods section.

EDITOR: L139 this drought has not been mentioned before and it should be characterized.

AUTHORS: The drought is now characterized in the introduction.

EDITOR: L151 adding a couple of words would enhance the flow of this sentence.

AUTHORS: A few words were added to improve this sentence. It now reads "Soil volumetric water content (VWC) data were collected after 2015 by reflectometers, with one Decagon Devices 5TM sensor in each plot measuring shallowly at 5 cm deep and three Campbell Scientific CS616 sensors in each plot collecting profiles from the surface to 15 cm deep."

EDITOR: L162 5 m seems low compared to the typical height of a mature canopy, i.e., >30 m.

AUTHORS: Added the word "low canopy" throughout the manuscript to make it clear that we are looking at the low canopy leaves, not the average canopy leaf.

EDITOR: L182-185 this sentence is not clear to me.

AUTHORS: This sentence was reworked to be two sentences, and the sentences around it were also reworked. It now reads "The Beer-Lambert law (Monsi, 1953) was used to convert the LAI data into solar radiation estimates, calculating the attenuation the canopy with a specific LAI invokes on the available (above-canopy) light. Annual patterns of photosynthetically active radiation (PAR) extinction coefficients are needed to calculate the attenuation given by the Beer-Lambert law. An annual pattern of these extinction coefficients was solved for by using two
years of data of the field-measured CTE2 control plot solar radiation, the tower weather station above-canopy solar radiation, and the MODIS LAI data. The three sets data were interpolated or averaged to daily values, and then the coefficients were calculated on the two years of data before the hurricane (so excluding the 2015 drought). These annual patterns were averaged and smoothed into one annual pattern of extinction coefficients that was applied for every year of the MODIS data."

EDITOR: L179 ter Steege 2018. Given the timing of the experiment this cannot be the primary reference of the method.

AUTHORS: This is the reference for the processing of the photos taken for the experiment, not for the taking of the photos. We added a few words to the beginning of the sentence to make this clearer: "In the reanalysis of the CTE1 data presented here, canopy photos were converted to global solar radiation data with a modified version of the Hemiphot method (ter Steege, 2018) as follows."

EDITOR: L211 how was the "uniform overcast sky factor" determined?

AUTHORS: A reference was added, it is an empirical calculation assuming uniform light over the sky. We changed this sentence for clarity to: "Underneath the canopy, PAR can be approximated as the sum of the direct light through
all open parts of the canopy and the diffuse light multiplied by 15% (based on empirical equations; Gates, 2012)."

AUTHORS: Agreed, alternations sounds much better.

AUTHORS: Added (214 to 919%).

AUTHORS: Yes, should have been "concluded previously"

AUTHORS: Added "air".

AUTHORS: Changed to "as seen by the large magnitude decreases in summer 2015 and the recovery afterwards in air relative humidity, soil moisture, and leaf saturation (Figure 2)."

AUTHORS: Yes, fixed this to "This will create"

AUTHORS: Satellite data scaling issues are now first discussed as a result in the last paragraph of the results. We expanded the discussion paragraph on the satellite data, acknowledging that the interesting thing is the quick recover and adding some discussion around this.

RC1: Van Beusekom et al. present measurements of the forest abiotic environment following experimental and natural disturbances in the Luquillo forest in Costa Rica [sic, should be Puerto Rico] over a period of 16 years. They use this information to assess the recovery time of different variables. Measurements such as these can provide valuable insights into the mechanisms which govern a particular ecosystem response – particularly when combined with measurements or modelling of plant responses. The paper is clearly written and presented, and the measurements are well-described and, as best as I can judge, appropriately controlled for changes in measurement technique. However, the key to the story of the paper is the definition of recovery time, and this appears to be somewhat arbitrarily defined with significant consequences for the results. On this basis, I cannot recommend the paper for publication in its current form.

Recovery time is defined in the paper as the point when the treated data time series crosses the control data time series and afterwards stays within 15

The choice of x and y is also critical, however. x=15

Even if one just eyeballs the plots, whilst one can be fairly confident about recovery for solar radiation, for throughfall it is much less clear (there is even divergence in 2014 following the supposed point of recovery, making it questionable whether recovery had even occurred). The definition of recovery time therefore needs some careful thought and sensitivity testing to give confident that the results are robust to the method used.

AUTHORS: We added a subsection to the methods "Quantifying Abiotic Interaction and Response" to explain the reasoning behind the recovery period metric and to make it clear that we ran sensitivity tests to find this method acceptable. The sensitivity testing has a subsection in the results, "Sensitivity Testing on Calculated Recovery Times". In the Introduction, we agree that recovery is an arbitrarily defined point, saying "This study attempts to quantify abiotic response as acute changes from a hurricane disturbance (experimental or otherwise) and recovery from the changes, for primary and secondary factors. Quantifying the responses makes it possible to assess if the experimental trimming data and satellite data are reasonable sources for studying the effect of hurricane disturbance and appear to be measuring the same abiotic system, as well as appreciate if different events cause substantially different responses. This study does not attempt to determine what amount of recovery is considered 'normal' conditions to biotic life, or in other words what would affect tertiary factors, but instead quantifies changes in the abiotic factors that can be used to frame the changes found in biotic factors post-hurricane in many previous studies including those of biotic abundance (Shiels et al., 2015), soil biochemistry (Arroyo and Silver, 2018), and plant reproduction (Zimmerman et al., 2018)." We have also added annual averages to the plot at the suggestion of the editor, and we discuss how the response seen in the annual averages compares with the metrics off the smoothed time series.

We also added some discussion around the purpose and intent of the recovery period metric, to clarify what "recovered" actually means. At the end of the Discussion, we now say : "The results in the sensitivity tests showed that quantifying recovery times using smoothed time series to homogenize data from several sources was a worthwhile effort, in that the abiotic factors can be sorted into quicker and slower recoveries, with results robust to the smoothing method. However, the definition of the 'recovered point' in time will be dependent on what biotic
life considers 'normal', necessarily different for every organism. Across all abiotic factors, this study used a uniform buffer metric of 'within 15% agreement between control and treated plots' once the experimental response is finished, in order to quantify the length of abiotic recovery as a starting point to for other researchers to frame the changes found in biotic factors post-hurricane. The percentage of the buffer did not matter to the results in most cases, as shown in the sensitivity tests. However, the percentage did alter the results on factors with more
complicated recovery paths, such as litter saturation (Figure 2e), and factors with more data variance, such as solar radiation and throughfall (Figures 1a, b). This points to the difficulty of quantifying recovery in an environmental system."

RC1: Minor comments:

Line 94. Were Campbell sensors used after 2015 as well? In the previous paragraph it indicates not, but here that they were.

AUTHORS: Campbell temperature 107 sensors were only used before 2015, and after 2015, soil VWC was
measured by CS616 sensors, which are also made by Campbell. We changed the wording in this paragraph to refer to the sensors as '107 sensors' instead of Campbell sensors to avoid this confusion.

RC1: L187. Is this really resilience? There is presumably just less vegetation to be disturbed, which naturally leads
to a smaller fluctuation. I would argue it just leads to lower amplitude of variability.

AUTHORS: We clarified this statement, pointing out that the treated plots are closer to recovery after the hurricane than the control plots are. "The passage of hurricane María, 2.8 years after the second experiment, happened when most of the abiotic factors had not recovered and the rest had just recovered. Temperatures after CTE1 and CTE2 and relative humidity in the air after CTE2 recovered in year three, less than half the time it took solar radiation to recover after CTE1, and a less than a third of the time it took throughfall to recover (Table 1). The effect of Hurricane María was smaller on the treated plots than the control plots, such that the absolute level of abiotic disturbance on the treated plots was smaller than on the control plots (Figures 1, 2). It is expected that the abiotic fluctuations from the hurricane would be smaller in the unrecovered treated plots than in the control plots since there is less vegetation to disturb. The fluctuation is smaller, but furthermore for most of the abiotic factors, the treated plots are closer to the recovered state after the hurricane than are the control plots. For example, there is more solar radiation reaching the forest floor in the treated plots than in the control plots before hurricane María, but after the hurricane there is less solar radiation reaching the forest floor in the treated plots than in the control plots (Figure 1a). The same scenario can be seen in the throughfall (Figure 1b), the temperatures to a lesser extent (Figures 1c, d), the soil moisture profile (Figure 2c), and the litter saturation (Figure 2e). The air relative humidity has the opposite scenario, showing treated plots closer to the recovery state of less humidity in the air after the hurricane (Figure 2a)."

RC1: L188. "greater disturbance" is not clear. Perhaps, "greater fluctuations in the measured abiotic variables due to disturbance"?

AUTHORS: Changed to "larger abiotic fluctuations due to disturbance".

RC1: L190. What exactly does it mean that "tree demographics were . . . dynamic"? Does this refer to the mix of ages in the forest, the rate of growth, the rate of turnover?

AUTHORS: All of the above. We added an explanation "(the rates of species and stem mortality and growth)"

This study integrated the observations both from in-situ and satellite platform for studying the dynamics of vegetation change in Luquillo Experimental Forest. Two canopy trimming experiments, one in 2004 and another in 2015, were designed as control experiments to reveal the vegetation recovery in response to the wind damage to the trees, especially for the case caused by the tropical storms (Irma and Maria) in 2017. The authors reported long term and continues time series of under-canopy solar radiation, throughfall, air temperature (under and above), soil water, and relative humidity and leaf saturation in the manuscript. This work can provide an insight into the vegetation recovery due to the wind disturbance in the tropical climate zone. However, the structure of the manuscript and approach for analysis the data are a bit confusing. I suggested that the authors provide a general review of the vegetation recovery in the introduction section and try to focus on the study results for the tropics. Here, I provided a few studies (listed in the reference) including observation and modelling works which are relevant for providing a general review of the wind disturbance research. The introduction of the canopy trimming experiment can move to the methodology section which can be the design of canopy trimming and natural disturbance events. Along with this discussion, the method applied for this study to identify the recovery period is questionable, and the authors didn't include or calculate the uncertainty caused by instruments, sampling approaches, or data analysis (smoothing). Regarding the issue for identifying the recovery period, I recommended the authors to analysis the annual maximum observations, for example the study made by Lin et al. (2016). By comparing annual maximum values over a long-term time series is helpful to identify the status of vegetation recovery period. I had several specific comments for the authors to improve the current version of this manuscript.

AUTHORS: Thank you for your detailed comments. We address those below. We moved the description of the experiment to the methods and we have also added the Mitchell (2013) reference in the introduction. We have greatly expanded the methodology description of the recovery metrics (with added results), citing Lin et al. (2016) as discussed under the comment 8. We also added annual averages to the figures and discussed how the annual summaries compare to the smoothed time series metrics in abiotic acute change and recovery time. The papers on windthrow modelling and tree mortality do not seem to be on topic as we are concerned with the abiotic environment and not the geographical extent of the disturbance.

RC2: Specific comments

1.Using the measurement of wetness of litter leaves and soil water to understand the canopy recovery physically is not reasonable. Although the wetness of litter leaves and soil moisture can be affected by the coverage of the over-story canopy, the magnitude of soil moisture and litter leaves are fixed which might only depend only on the soil property and leaf types. Please explain how to use the observation of soil moisture and wetness of litter leaves to reveal the status vegetation recovery.

AUTHORS: We are not attempting to understand canopy recovery, but instead how the forest abiotic environment responds to the vegetation changes, and when the abiotic environment is recovered to its pre-hurricane state. We added this sentence in the introduction "Instead of trying to estimate if and when the vegetation has returned to its pre-disturbance state, insight on ecosystem health can be gained by studying how the abiotic factors respond to the disturbance." We agree that the timeline of soil and litter moisture recovery very much depends on the types of soil and leaves involved. We are focused on the response, not the specific timeline. This comment has been added to the discussion in the section talking about the soil and litter patterns: "Specific timelines for recovery would be expected to be highly influenced by the tree species and soil types, and the rates seen here for all abiotic factors would not necessarily apply to all hurricane-effected tropical forests. Nevertheless, general patterns might be expected to hold." We have pointed out that similar patterns to the litter saturation response patterns seen here were also presented in Southeastern United States.

RC2: 2.P2L61: (wetness of canopy and litter leaves) How to determine the wetness of canopy leave and litter leaves.

AUTHORS: This is discussed in the methods, "Leaf saturation data were collected after 2015 by Decagon Devices dielectric leaf wetness sensors in the canopy leaves 5 m up from the ground and in the litter leaf layer."

RC2: 3.P3L79: "locally to the points", Can you show the original points in your results?

AUTHORS: We added points to the plots, but on suggestion of the editor we needed to make the plots clearer, so we have removed the points and added annual averaging lines instead. The tracked changes show the point plots; hopefully you agree that the annual average lines are much clearer than the point plots.

RC2: 4.P3L90: The MODIS only measured the sink temperature of the surface. Why did the authors compare the air temperature observations to the MODIS LST observations?

AUTHORS: We compared to see if any of the forest cover change seen in the field observations of temperature (above and below canopy) were comparable to the LST. We clarified this in the methods, saying "MODIS LST measures energy balance at the land surface, so is not representative of air temperature under the canopy but it will be affected by changes in air temperature. Annual maximums of LST and air temperature are highly correlated across the globe with correlation strongest in forested areas (Mildrexler et al., 2011), and LST has been shown to respond to forest cover changes in other areas of the tropics (van Leeuwen et al., 2011)." Again in the results, we clarified "The (MODIS LST-estimated) temperature satellite data plot between the field air temperature data measured below the canopy and that measured above the canopy at 30 m (black and gray lines respectively, Figure 1c), giving evidence that the satellite measurements were affected by a vertically averaged Earth, as might be expected from a LST representative of surface energy balance."

RC2: 5.P3L92: How many 5TM sensors were deployed for soil water observation? What is the minimum requirement for avoiding the spatial heterogeneity under canopy at this study site?

AUTHORS: We use several sensors in each of the 6 plots to avoid this problem. We added the number of sensors to each paragraph in the methods, for each type of sensors. At the beginning of the methods, we now explain "To account for spatial heterogeneity under the canopy, multiple sensors were used in each plot were used and the results were averaged in all control and treated plots (with quality control)."

RC2: 6.P4L115-L124: Too many details were lost or cannot be found. For example, the relationship between the 8-day MODIS LAI and 8-day in-situ solar radiation was built up for converting the MODIS LAI to solar radiation for the study site, but the authors didn't present this information and uncertainty.

AUTHORS: We clarified this section by expanding description to "The Beer-Lambert law (Monsi, 1953) was used to convert the LAI data into solar radiation estimates, calculating the attenuation the canopy with a specific LAI invokes on the available (above-canopy) light. Annual patterns of photosynthetically active radiation (PAR) extinction coefficients are needed to calculate the attenuation given by the Beer-Lambert law. An annual pattern of these extinction coefficients was solved for by using two years of data of the field-measured CTE2 control plot solar radiation, the tower weather station above-canopy solar radiation, and the MODIS LAI data. The three sets data were interpolated or averaged to daily values, and then the coefficients were calculated on the two years of data before the hurricane (so excluding the 2015 drought). These annual patterns were averaged and smoothed into one annual pattern of extinction coefficients that was applied for every year of the MODIS data.." We also now separated each variable's method of processing into it's own subsection, for clarity.

RC2: 7.P5L149-150: The reason for applying 1year smooth window is not clear, please explain in the method section.

AUTHORS: We added an expanded explanation to the methods: "The LOESS degree of smoothing is contingent on the size of the local neighborhood, which here was always chosen to be one year of data around each point. The yearly smoothing was done to extract the larger signal from the data and to homogenize the different collection intervals of the data. The automated sensor field data captured larger amounts of background noise than the temporally smoothed rain funnel data and the geographically smoothed satellite data; and to a lesser extent, the geographically-smoothed soil sample, litterbag, and canopy photo data. The one-year smoothing neighborhood was chosen to be longer than the longest length of time between repeat measurements across all data types and methods."

RC2: 8.P5L159-161: The way for justifying the recovery period is not clear, please explain the method in detail.

AUTHORS: We add a subsection to the methods "Quantifying Abiotic Interaction and Response" to explain the reasoning behind the recovery period metric and to make it clear that we ran sensitivity tests to find this method acceptable. The sensitivity testing has a subsection in the results, "Sensitivity Testing on Calculated Recovery Times".

We also added some discussion around the purpose and intent of the recovery period metric, to clarify what "recovered" actually means. At the end of the Discussion, we now say : "The results in the sensitivity tests showed that quantifying recovery times using smoothed time series to homogenize data from several sources was a worthwhile effort, in that the abiotic factors can be sorted into quicker and slower recoveries, with results robust to the smoothing method. However, the definition of the 'recovered point' in time will be dependent on what biotic life considers 'normal', necessarily different for every organism. Across all abiotic factors, this study used a uniform buffer metric of 'within 15% agreement between control and treated plots' once the experimental response is finished, in order to quantify the length of abiotic recovery as a starting point to for other researchers to frame the changes found in biotic factors post-hurricane. The percentage of the buffer did not matter to the results in most cases, as shown in the sensitivity tests. However, the percentage did alter the results on factors with more complicated recovery paths, such as litter saturation (Figure 2e), and factors with more data variance, such as solar radiation and throughfall (Figures 1a, b). This points to the difficulty of quantifying recovery in an environmental system."

RC2: 9.P6L169-176: I didn't understand why the authors reported the residuals of the statistical analysis in this paragraph. Is this information helpful for understanding the uncertainty of various measurements?

AUTHORS: This is the method for correlating time series: we remove seasonality and trends and correlate what is left over. We had called these 'leftovers' as 'residuals. To make this clearer, we changed the words "residuals of [data]" to "the prewhitened [data]".

RC2: 10.In the Discussion section: It is very difficult for me to find/justify the information of the recovery periods, such 10 years, 2.8 years and others values from Figs 1 and 2. I recommended the authors to indicate such a piece of information both in this section and key Figs.

AUTHORS: We have redone the figures to have thick lines of annual averages to aid in the visualization of the recovery, and also added circles marking the recovery point (as well as bars marking the acute changes). We also added a few more references to Table 1, where all the specific recovery information is at.

| Page 8: [1] Deleted | Ashley Van Beusekom2 | 2/4/20 11:12:00 PM |
|---|---|---|
| Page 11: [2] Deleted | Ashley Van Beusekom2 | 2/20/20 12:18:00 AM |
| Page 11: [3] Deleted | Ashley Van Beusekom2 | 2/4/20 11:37:00 PM |
| Page 11: [4] Deleted | Ashley Van Beusekom2 | 2/20/20 12:07:00 AM |
| Page 11: [5] Deleted | Ashley Van Beusekom2 | 2/20/20 12:11:00 AM |
| Page 19: [6] Deleted | Ashley Van Beusekom | 12/2/19 12:27:00 PM |
| Page 19: [7] Deleted | Ashley Van Beusekom2 | 2/18/20 10:13:00 PM |
| Page 19: [7] Deleted | Ashley Van Beusekom2 | 2/18/20 10:13:00 PM |
| Page 19: [8] Deleted | Ashley Van Beusekom2 | 2/18/20 10:13:00 PM |
| Page 19: [8] Deleted | Ashley Van Beusekom2 | 2/18/20 10:13:00 PM |
| Page 19: [8] Deleted | Ashley Van Beusekom2 | 2/18/20 10:13:00 PM |
| Page 19: [8] Deleted | Ashley Van Beusekom2 | 2/18/20 10:13:00 PM |
| Page 19: [8] Deleted | Ashley Van Beusekom2 | 2/18/20 10:13:00 PM |
| Page 19: [8] Deleted | Ashley Van Beusekom2 | 2/18/20 10:13:00 PM |
| Page 19: [8] Deleted | Ashley Van Beusekom2 | 2/18/20 10:13:00 PM |
| Page 19: [8] Deleted | Ashley Van Beusekom2 | 2/18/20 10:13:00 PM |

| Page 19: [8] Deleted | Ashley Van Beusekom2 | 2/18/20 10:13:00 PM |

| Page 20: [9] Deleted | Ashley Van Beusekom2 | 2/18/20 10:14:00 PM |

| Page 20: [9] Deleted | Ashley Van Beusekom2 | 2/18/20 10:14:00 PM |

| Page 20: [10] Deleted | Ashley Van Beusekom2 | 2/18/20 10:14:00 PM |

| Page 20: [10] Deleted | Ashley Van Beusekom2 | 2/18/20 10:14:00 PM |

| Page 20: [10] Deleted | Ashley Van Beusekom2 | 2/18/20 10:14:00 PM |

| Page 20: [10] Deleted | Ashley Van Beusekom2 | 2/18/20 10:14:00 PM |

| Page 20: [10] Deleted | Ashley Van Beusekom2 | 2/18/20 10:14:00 PM |

---

## Author Response (AR3)

[revised manuscript text omitted]

\* First column is percentage change from control; second col...
\*\* First column is percentage change from before hurricane M...

[Figure]

Surface colors: Smooth-mean data   Mean data   Daily data

Figure 3: Sensitivity studies on primary factor calculated recovery times. Calculated years until recovery for each factor are plotted on the vertical axis, with variables of buffers on the other axes. The buffers are the required fraction of the acute change the difference in control and treated plots must approach, on the recovery day for initial closeness (init) and for 6 months after for post small-differences (post). Plot column one is for after canopy experiment (CTE) 1 and column 2 is for after CTE2. Surface color represents data summary methods as indicated, and absence of values for specified buffer conditions means the factor did not recover during the time period of the experiment with the specified buffers. Plots show a) solar radiation beneath the canopy; b) throughfall; c) air temperature; and d) soil temperature.

**CTE1 Years until Recovery**    **CTE2 Years until Recovery**

[Figure]

**a**

**b**

**c**

**d**

**e**

**Surface colors:** Smooth-mean data  Smooth data  Daily data

**Figure 4: Sensitivity studies on secondary factor calculated recovery times. All markings are the same as in Figure 3. Plots show a) air relative humidity; b) soil moisture shallow; c) soil moisture profile; d) low canopy leaf saturation; and e) litter leaf saturation.**

780

785

795

790 times are robust and reflect the ecological recovery of a biophysical process rather than simply reflecting the assumptions made when defining "recovery time" in this study. I would, therefore, like to encourage the authors to evaluate and report the impact of alternative definitions on the recovery times. This analysis should help to convince the readers that the recovery times of different biophysical processes are indeed different or contrary that these differences disappear when different definitions are used.

AUTHORS: Thank you for your continued comments. We have greatly expanded the sensitivity studies in the methods and in the results, including 2 figures (Figures 3 and 4) showing the impact of differing data summarization methods and different metrics for recovery of the treated data back to the level of the control data. The analysis shows that the recovery times of the different biophysical processes can become the same if we go the extremes of

800 the definitions of recovery (with smallest or largest buffers), but using moderate buffers, the recovery times are distinguishable among the processes. We have taken the reviewers' most recent suggestions on how to make the sensitivity study and the response time definition better and have included them in the paper. The specifics of these are discussed below in the responses to the reviewers.

805

The authors have made substantial textual revisions to the manuscript. The methods are much better described than previously. Regarding my main point in my original review of the definition of recovery time, I was disappointed to see that the authors edited away most of my comments before responding. E.g. there is no response to the critique of the crossing requirement, or the differences between treated and control pre-treatment, or the difference in

The authors have retained the same definition of recovery time basing the justification on visual interpretation. I am still of the opinion that this justification is weak. Throughfall is a clear example why the visual recovery is potentially misleading. It is indeed true that ca. 8 years after trimming the two timeseries appear to sit on top of each other. However, prior to the first trimming experiment, the control and treated timeseries show a notable offset. There is therefore no evidence that the control and treated timeseries should have the same throughfall in a "recovered" state and that the close correspondence after 8 years is anything other than an aberration (the lines also diverge again in 2014!). Arguably, throughfall could be considered recovered after ca. 5 years based on the 15% threshold alone, with no crossing requirement. I thoroughly agree with eyeballing data to make sure that statistics make sense; but, putting eyeballing ahead of good statistical justification for the recovery time appears a bit about-face.

The authors have added a sensitivity study of the effects of changing different parameters of their recovery definition and this is useful and does go some way to indicating the sensitivity of different variables to the definition. I would never expect an exhaustive review of different definitions of recovery time as that is not the point of this manuscript, but somehow I think that as this manuscript is all about recovery time, the definition of recovery time is a key point that still warrants more careful treatment (for instance, using a basket of methods). Otherwise, how can we have confidence in the relative times for recovery of the different variables?

The ranges for the sensitivity analysis would be easier to interpret if presented in a table (or in a figure, e.g. error bars on a bar plot of recovery time).

AUTHORS: Thank you for your continued review. I think there must have been an error in the upload of the initial comments, because it seems your comments on "the critique of the crossing requirement, or the differences between treated and control pre-treatment, or the difference in strictness of a 15% threshold for different variables" finished mid-sentence, so we did not see the whole comment; sorry for the misunderstanding. We appreciate the idea of a bar plot with the results of the sensitivity study; it seems this will make the results much clearer and clarify that sensitivity studies were run if we include some figures. To this end, we have added a bracket of methods (recovery based on linear representation of raw data, recovery based on smoothed polynomial representation of raw data, varying buffers) and included their results in 2 new figures (3 and 4) as well as added ranges in the Table 1. We also added a paragraph on how the sensitivity study shows the shortcomings of using the strict quantitative recovery metric with the data (lines 347-350). This includes a discussion of throughfall results similar to what you have

pointed out. We changed the discussion to refer to the throughfall recovery with more uncertainty (see lines 483-485), as well as the abstract (line 16).

Line 733. "hurricane-affected"
AUTHORS: Fixed.

Line 777. "starting point for"
AUTHORS: Fixed.

RC2:
Thanks for preparing the revised manuscript of this study and I have been gone through all sections in the revised manuscript. The quality of this manuscript has been improved. I recommended the authors to add a sensitivity analysis for quantifying the response to the treatments and hurricanes (section 2.2). This type of analysis will help support the results/conclusion made by the authors in this study. For example, the authors set a buffer size by adding 15% to find overlapping as the start of the recovery state. How about using different values to find overlaying? In this approach, using different buffer sizes will result in different recovery periods and also can define different periods before the next disturbance. I thus suggested the authors not only using a single value to define the state of recovery but using a relative comparison between two periods/states to choose the buffer size. By doing so, the authors can report the uncertainty of using different buffer sizes to calculate the recovery period.

AUTHORS: Thank you for your continued review. We have added a bracket of methods (recovery based on linear representation of raw data, recovery based on smoothed polynomial representation of raw data, varying buffers) and included their results in 2 new figures (3 and 4) as well as added ranges to the Table 1. We appreciate the idea of using a relative comparison between two states to choose the buffer size and have now used this method as detailed in Equation 4. We agree that a relative buffer makes more sense given the noise in the data, and have discussed this in lines 339-346.

Please adjust the list order of the "van Leeuwen ..." in the reference list.

AUTHORS: Since this person keeps the prefix of their name lowercase in the proper Dutch manner, I think they get alphabetized with the L's. My name is a capital "Van", Americanized, so I am alphabetized with the V's. I prefer to let the copy editor decide where to put that entry.